# PDZD7-MYO7A complex identified in enriched stereocilia membranes

Clive P Morgan[1†], Jocelyn F Krey[1†], M'hamed Grati[2], Bo Zhao[3], Shannon Fallen[1], Abhiraami Kannan-Sundhari[2], Xue Zhong Liu[2], Dongseok Choi[4,5], Ulrich Müller[3], Peter G Barr-Gillespie[1*]

[1]Oregon Hearing Research Center and Vollum Institute, Oregon Health and Science University, Portland, United States; [2]Department of Otolaryngology, Miller School of Medicine, University of Miami, Miami, United States; [3]Dorris Neuroscience Center, The Scripps Research Institute, La Jolla, United States; [4]OHSU-PSU School of Public Health, Oregon Health and Science University, Portland, United States; [5]Graduate School of Dentistry, Kyung Hee University, Seoul, Korea

**Abstract** While more than 70 genes have been linked to deafness, most of which are expressed in mechanosensory hair cells of the inner ear, a challenge has been to link these genes into molecular pathways. One example is *Myo7a* (myosin VIIA), in which deafness mutations affect the development and function of the mechanically sensitive stereocilia of hair cells. We describe here a procedure for the isolation of low-abundance protein complexes from stereocilia membrane fractions. Using this procedure, combined with identification and quantitation of proteins with mass spectrometry, we demonstrate that MYO7A forms a complex with PDZD7, a paralog of USH1C and DFNB31. MYO7A and PDZD7 interact in tissue-culture cells, and co-localize to the ankle-link region of stereocilia in wild-type but not *Myo7a* mutant mice. Our data thus describe a new paradigm for the interrogation of low-abundance protein complexes in hair cell stereocilia and establish an unanticipated link between MYO7A and PDZD7.

*For correspondence: gillespp@ohsu.edu

†These authors contributed equally to this work

Competing interests: The authors declare that no competing interests exist.

## Introduction

Hair cells are the sensory cells of the inner ear; they transduce mechanical signals evoked by sound and head movement and relay these signals to the central nervous system. The mechanically sensitive organelle responsible for mechanotransduction is the hair bundle, a cluster of ~100 actin-filled stereocilia that protrude apically from a hair cell (*Gillespie and Müller, 2009*). Because stereocilia bend at their bases yet are coupled together with elastic linkages, the bundle pivots as a whole when exposed to mechanical stimuli. One special type of linkage, the tip link, couples bundle pivoting to electrical excitation of the hair cell. The tip link is composed of a dimer of CDH23 (cadherin 23) molecules interacting with a dimer of PCDH15 (protocadherin 15) molecules (*Kazmierczak et al., 2007*), and is coupled to the mechanically sensitive transduction channel. Another class of linkage, the ankle link, couples stereocilia together at their bases, transiently in auditory hair cells but persistently in vestibular hair cells (*Goodyear and Richardson, 1999*; *Adato et al., 2005*; *McGee et al., 2006*; *Michalski et al., 2007*). Because many genes encoding proteins known to be associated with these links are affected by mutations that cause deafness, it is of great interest how these links couple with proteins within the stereocilia.

Myosin VIIA (MYO7A) was the first protein involved in hair-cell mechanotransduction to be identified by genetics (*Gibson et al., 1995*; *Weil et al., 1996*). While MYO7A has been implicated in ankle-link positioning by its location in frog stereocilia (*Hasson et al., 1997*) and by mislocalization

**eLife digest** Inside the inner ear, sensory cells called hair cells detect and respond to sounds and head movements. These cells have a mechanically sensitive structure called the hair bundle, which is made of many thin projections called stereocilia. The stereocilia are linked so that when they bend in response to a sound or head movement, the whole hair bundle moves as one.

A protein called myosin VIIA (MYO7A) is thought to be involved in forming links at the base of stereocilia (so-called 'ankle links') and relaying signals from the stereocilia to the rest of the hair cell. However, it is not known how MYO7A interacts with the proteins that make up the ankle links.

To address this question. Morgan, Krey et al. developed a new method for isolating groups of proteins from the inner ear of chick embryos that are only found in low quantities. Using this method, it was possible to isolate MYO7A along with other proteins it associates with. One of these proteins – called PDZD7 – is known to be part of ankle links. The next step following on from this work is to use this new method to study other important groups of proteins that are even more scarce in hair bundles.

of ankle-link proteins in *Myo7a*-null mice (*Michalski et al., 2007*), more precise localization experiments suggested that MYO7A also participates in mechanotransduction (*Grati and Kachar, 2011*).

These differing roles could be reconciled if MYO7A forms several different protein complexes. The tail of MYO7A has two tandem FERM-MyTH4 domains and one SH3 domain, each of which might interact with stereocilia proteins. Biochemical and structural studies have indicated that MYO7A interacts both with PCDH15 (*Senften et al., 2006*) and with a complex including USH1C, USH1G, and CDH23 (*Boëda et al., 2002*; *Wu et al., 2011*). Preliminary evidence suggests that MYO7A interacts with ADGRV1, which was formerly known as GPR98 and VLGR1 (*Hamann et al., 2015*), USH2A (usherin), and DFNB31 (whirlin) (*Delprat et al., 2005*; *Michalski et al., 2007*). No interactions of MYO7A have been reported with PDZD7, another member of the ankle-link complex (*Grati et al., 2012*; *Zou et al., 2014*), and how MYO7A positions the ankle-link complex remains unclear.

While genetic methods have been successful in identifying essential stereocilia proteins, the approach can miss proteins that are more widely expressed or that can be compensated for by the expression of a paralog. To determine all the proteins present in hair bundles, we have developed biochemical methods for their characterization, coupling purification of bundles using the twist-off method with detection of proteins by mass spectrometry (*Gillespie and Hudspeth, 1991*; *Shin et al., 2007*, *2013*). Because of the fine dissection and tissue manipulation required, however, the twist-off method is limited in its throughput and is less ideal for biochemical experiments examining rare protein-protein complexes in bundles.

We therefore developed a high-throughput method for stereocilia-membrane enrichment using the monoclonal antibody D10, which allowed us to prepare large amounts of solubilized inner-ear material that is suitable for immunoaffinity purification of protein complexes from stereocilia detergent extracts. We initially used this approach to determine which proteins interact with MYO7A in the inner ear, using the monoclonal antibody 138-1 to show that MYO7A interacts tightly with several scaffolding proteins, notably PDZD7. By immunoprecipitating expressed proteins in tissue-culture cells, we confirmed that the interaction of PDZD7 and MYO7A is direct. Moreover, MYO7A is also in a protein complex including ADGRV1, which likely also includes PDZD7. We also show that USH1C, USH1G, and CDH23 co-precipitate from chick stereocilia with MYO7A, confirming the presence in the tissue of a complex that had only been suggested by genetics and recombinant-protein expression. Because other scaffolding proteins co-precipitate with MYO7A, these experiments show not only that MYO7A contributes to the tip-link and ankle-link complexes, but also suggest that MYO7A participates in other stereocilia protein complexes. Finally, the experiments indicate that the coupled stereocilia-membrane purification and immunoaffinity isolation methods will be useful for isolating even very rare protein complexes from stereocilia.

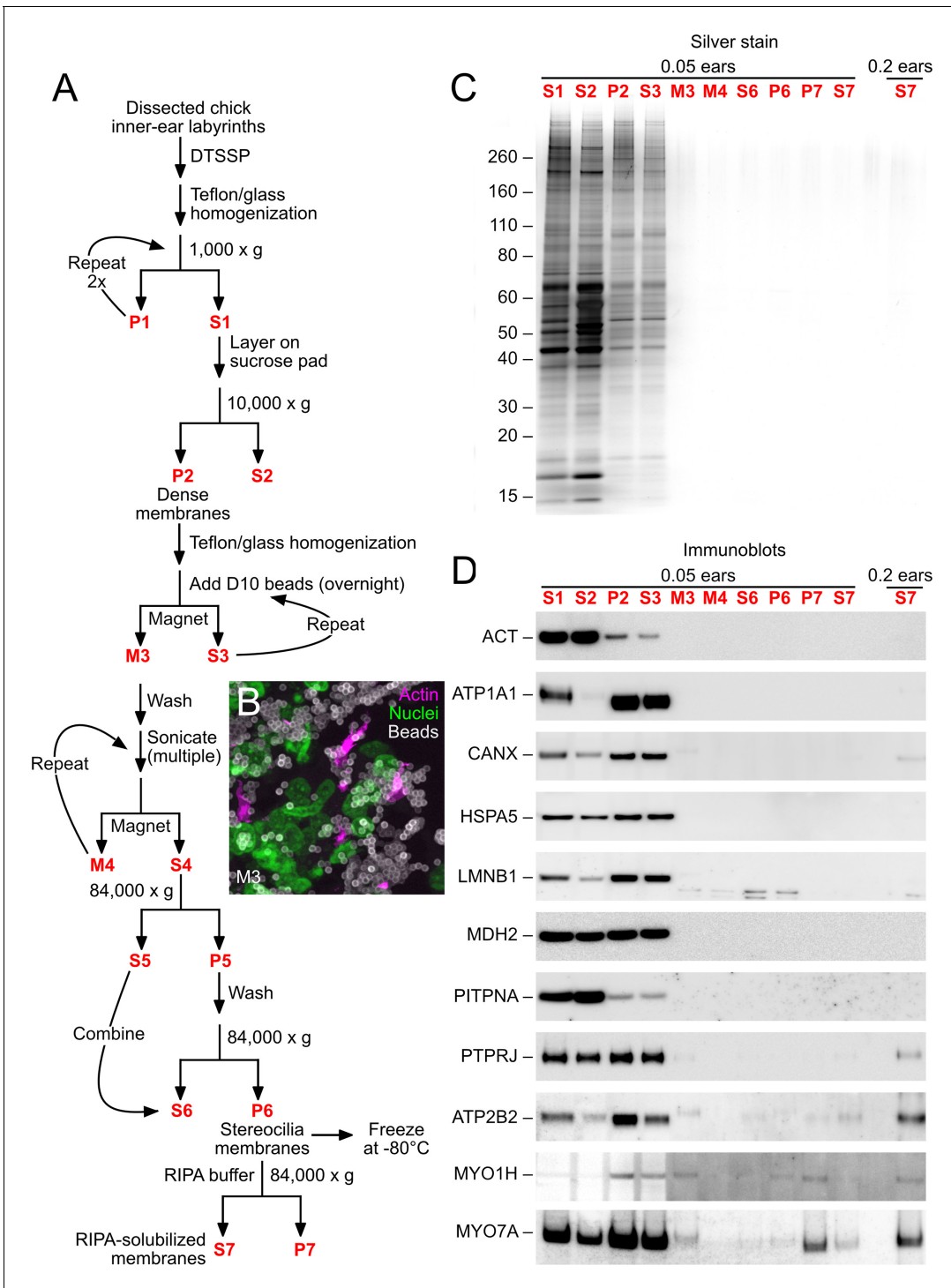

**Figure 1.** D10 chick stereocilia-membrane enrichment procedure. (**A**) Flow chart for stereocilia membrane enrichment. Bold red lettering highlights the principal steps of the procedure. (**B**) Imaging of M3 fraction of D10 beads; samples were labeled with phalloidin (magenta, for actin), DAPI (green, for nuclei), and anti-mouse IgG (white, for D10 antibody on beads). Aggregation of beads and binding of nuclei (and other contaminants) limits the enrichment gained with this step. (**C**) SDS-PAGE analysis of enrichment fractions; gel stained with silver. Note that fraction S7 was loaded in a second lane with 4x more material. (**D**) Immunoblot analysis of purification fractions. For actin (ACT), lanes 1–10 had 0.01 ear-equivalents loaded and lane 11 had 0.2 ears. For all other immunoblots, lanes 1–10 had 0.05 ear-equivalents loaded and lane 11 had 0.2 ears. Other than actin, all proteins are referred to by their official gene symbols.

The following figure supplement is available for figure 1:

*Figure 1 continued on next page*

*Figure 1 continued*

**Figure supplement 1.** Protein recovery with D10 and twist-off methods.

## Results

### D10 chick stereocilia membrane preparation

We enriched stereocilia membranes from chicks using the D10 preparation, named for the monoclonal antibody that is critical for the procedure (*Figure 1*). D10 recognizes PTPRQ (*Goodyear et al., 2003*), the most abundant transmembrane protein of hair bundles (*Shin et al., 2013*). As in our previous experiments coupling stereocilia isolation with mass spectrometry (*Shin et al., 2007, 2013*), we used E19-E21 chicks because of chicks' availability, moderate cost, sequenced genome, and near-mature inner ear. In most cases, we treated the exposed surfaces of dissected inner ears with DTSSP, a membrane-impermeable, reversible crosslinker with two N-hydroxysuccinimide groups coupled with an 12 Å spacer that includes a disulfide bond (*Staros, 1982, 1988*); this treatment stabilizes complexes of membrane proteins (*Corgiat et al., 2014*), presumably including ankle links and tip links. DTSSP treatment should not affect intracellular protein complexes, however, and is reversed by reduction. If their binding affinity is sufficiently high, cytoplasmic proteins in association with membrane proteins should still be detected, however, even without covalent crosslinking. Because one investigator can comfortably dissect only ~200 ears per day, to allow larger preparations ($\geq$1000 ears), we froze tissues daily after quenching the DTSSP reaction.

After thawing the inner-ear tissue, we used conventional biochemical procedures to enrich stereocilia membranes. We used antibodies to ATP2B2, the plasma-membrane $Ca^{2+}$ pump of stereocilia (*Dumont et al., 2001*), to locate fractions containing stereocilia membranes (*Figure 1D*). Tissues were homogenized, subjected to differential centrifugation to remove nuclei (yielding fraction S1; *Figure 1A*), and dense organelles were sedimented on top of a sucrose pad). In pilot experiments, we subjected the post-nuclear supernatant to fractionation using discontinuous equilibrium sucrose gradients; most of the ATP2B2 and hence the stereocilia membranes sedimented to the top of the 1.8 M sucrose layer. In large-scale experiments, we layered the post-nuclear supernatant on a cushion of 2.2 M sucrose, centrifuged to equilibrium, and collected material at the interface (fraction P2).

Stereocilia membranes were captured using magnetic beads coupled with D10 (fraction M3). We labeled beads with the antibody at high density, which facilitated efficient binding to stereocilia and enrichment with a magnet. Because stereocilia were present at extremely low concentrations in the extract, however, there was a large excess of antibody-coupled beads over stereocilia membranes. As a consequence, substantial amounts of non-stereocilia protein bound nonspecifically to the beads. While phalloidin staining showed that many of the D10 beads were decorated with stereocilia (*Figure 1B*), nuclei and other contaminants were also associated with the beads, presumably nonspecifically. Membranes were eluted using sonication (fraction S4); they were then washed several times prior to detergent extraction, yielding enriched stereocilia membranes (fraction P6). We solubilized protein complexes using RIPA buffer, a stringent buffer that contains both SDS and deoxycholate; the final solution, fraction S7, was used for subsequent immunoaffinity purification steps.

We examined the purification fractions using silver-stained SDS-PAGE gels (*Figure 1C*) and immunoblotting of marker proteins (*Figure 1D*). Very little total protein bound to the D10 beads (M3), and cytosolic (HSPA5, PITPNA1), nuclear (LMNB1), mitochondrial (MDH2), and plasma-membrane (ATP1A1) proteins were largely absent from D10 eluates (S4). A modest amount of the endoplasmic reticulum marker CANX was visible in the final RIPA-solubilized fraction (S7), however, as was the supporting-cell antigen PTPRJ. Nearly all of the ATP2B2 was detected in S7, however, showing that stereocilia membranes were efficiently solubilized.

In quantitative immunoblotting experiments, we found that the total amount of actin and ATP2B2 per ear, as compared to the amount in a single utricle, was much higher in the D10 stereocilia membrane preparation than in the twist-off hair-bundle preparation (*Figure 1—figure supplement 1*). This elevated level of stereocilia protein derived in part from sampling seven organs (cochlea, lagena, utricle, saccule, and three semicircular canals) rather than just one. Contaminating ATP1A1 from the basolateral membrane was detectable in the D10 preparation, but at modest levels

**Table 1.** Comparison of stereocilia purification methods. The D10 preparation enriches stereocilia membranes, and therefore is not directly comparable to the twist-off method, which isolates the entire hair bundle.

| Parameter | Twist-off | D10 |
|---|---|---|
| Throughput | <40 ears/person/day | >200 ears/person/day |
| Stereocilia purity | >90% | - |
| Recovery | ~1/3 | >1/2 |
| Organs sampled | 1 | 7 |
| Stereocilia protein recovered | 1x | 10x |
| Stereocilia protein per day | 1x | 50x |
| Subsequent purification | Challenging | Easy |

compared to that in one utricle (*Figure 1—figure supplement 1*). The D10 and twist-off preparations were compared in *Table 1*; note that a major advantage of the D10 preparation is its high throughput.

## Mass spectrometry analysis of stereocilia membrane preparation

We used shotgun (data-dependent acquisition, DDA) and targeted (parallel reaction monitoring, PRM) mass spectrometry to determine enrichment of key proteins in the stereocilia membrane preparation (*Figure 2A–E*). In fractions from two independent preparations, each of 1000 chick ears, the total amount of protein (*Table 2*) decreased from ~110 mg in S1 (post-nuclear supernatant) to ~1 mg in S7 (RIPA solubilized membranes), consistent with the silver-stain analysis. We analyzed two technical replicates each of the two preparations by shotgun mass spectrometry (four runs per fraction), and used Andromeda peptide searching and MaxQuant protein assembly and quantitation to determine the protein composition of each fraction. Equal amounts of protein were analyzed, and over 3000 proteins were detected in each fraction. Proteins were quantified using the intensity-based absolute quantification (iBAQ) method (*Schwanhäusser et al., 2011*). To estimate the molar fraction of each protein in each sample, we converted iBAQ to relative iBAQ (riBAQ), which is the iBAQ for a given protein divided by the summed iBAQ for all the proteins of the fraction after exclusion of contaminants (*Shin et al., 2013*; *Krey et al., 2014*). All data are deposited at ProteomeXchange (http://www.proteomexchange.org) with the identifier PXD004222, and the analysis is tabulated in *Figure 2—source data 1*.

Previous proteomics experiments indicated that PTPRQ and ATP2B2 are the most abundant membrane proteins of chick vestibular stereocilia (*Shin et al., 2013*); other notable membrane proteins enriched in stereocilia include SLC9A6 and SLC9A9 (*Hill et al., 2006a*); ATP8B1, NPTN, STARD10, and EFR3A (*Zhao et al., 2012*); CDH23, PTPRF, LGALS1, C2CD2L, KIAA1211, and KIAA1549 (*Shin et al., 2013*), as well as SLC34A2, a $Na^+$-$P_i$ transporter enriched in in cochlear stereocilia (*Avenarius et al., 2014*). With the exception of KIAA1211, which was not detected in the shotgun experiments, all of these stereocilia membrane proteins were enriched as the preparation proceeded or were found only in fraction S7 (*Figure 2D*).

To determine in an unbiased way whether stereocilia membranes increased in relative abundance, we calculated a slope for each protein of the preparation across the fractions where we expected enrichment (S1, P2, M3, P6, and S7). Of the 3036 proteins detected in at least four of the six fractions, PTPRQ had the largest 'enrichment slope' (*Figure 2A*). The MYO3A-MYO3B group, with a somewhat steeper slope, was only detected in three fractions. SLC34A2 was highly enriched, as was XIRP2; while the function of the latter protein is poorly understood, it does localize to the membrane-cytoskeleton interface in stereocilia and could bind membranes itself (*Francis et al., 2015*). Many other proteins enriched in the D10 preparation have annotated transmembrane domains, although some of these proteins likely derive from membranes other than those of stereocilia. Other proteins were known stereocilia proteins (ESPN, ANKRD24, MYO1H). Although ATP2B2 is part of a protein group also containing ATP2B1 and ATP2B4, which are not located in stereocilia (*Dumont et al., 2001*), we estimated ATP2B2's contribution to the ATP2B1/2/4 group riBAQ value

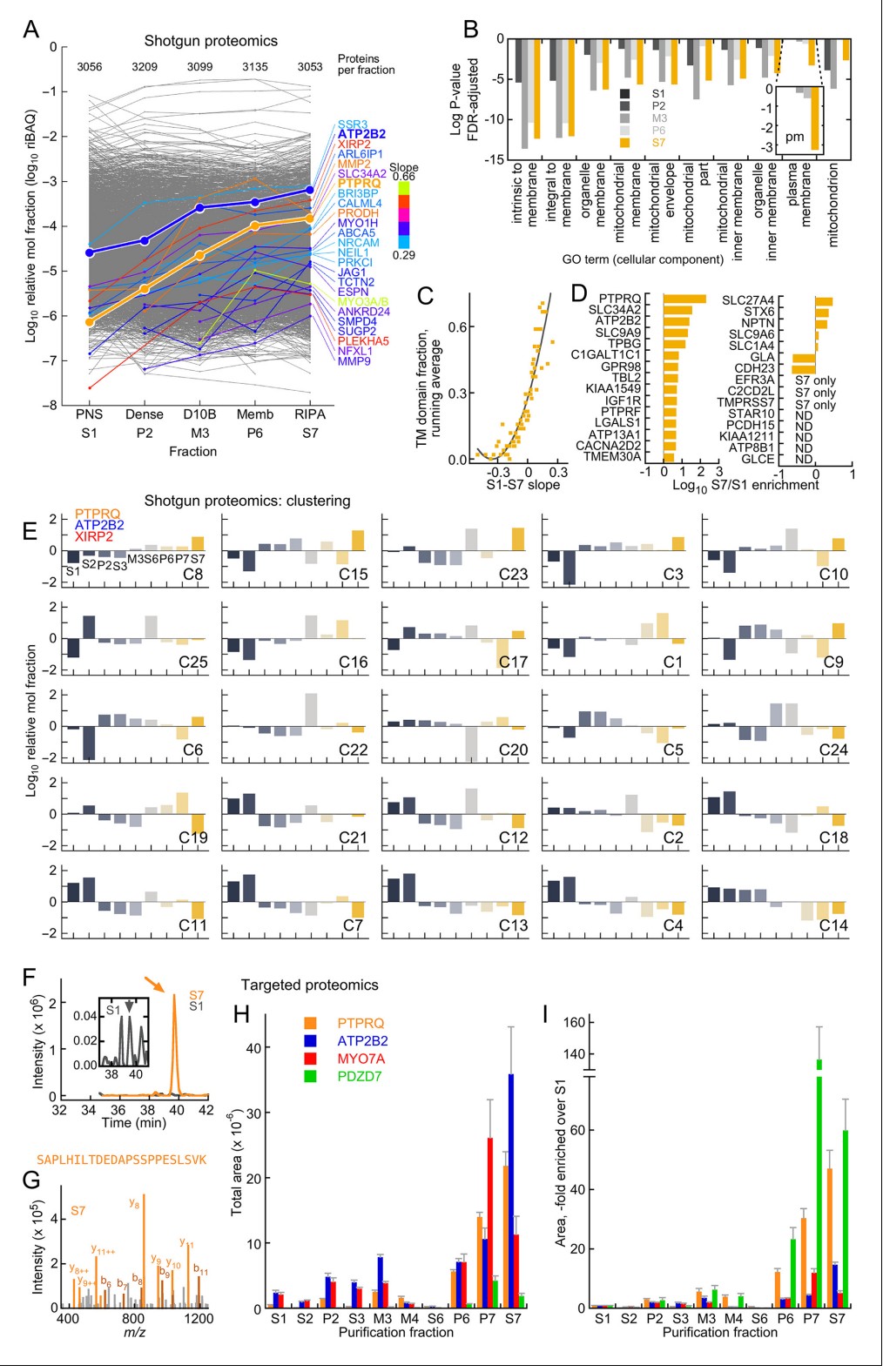

**Figure 2.** Mass spectrometry analysis of D10 stereocilia-membrane enrichment. (**A**) Shotgun mass-spectrometry quantitation of 3313 proteins detected in at least three of five purification fractions where enrichment of stereocilia membranes was expected. The slope of the purification fraction (assuming an interval of 1) vs. log riBAQ was calculated for each protein, and the top 25 proteins with the steepest positive slopes ('enrichment slope') were highlighted. Color indicates the slope steepness. For this analysis, ATP2B2 was split out of the group also

*Figure 2 continued on next page*

*Figure 2 continued*

containing ATP2B1 and ATP2B4 group; data corresponding both ATP2B2 and PTPRQ were emphasized to illustrate the effectiveness of the stereocilia-membrane enrichment. The number of proteins detected in each fraction was also indicated. (B) Gene ontology analysis with DAVID of the S7 (RIPA-soluble) fraction using S1 (PNS) as background. The top ten cellular component terms enriched are indicated; the log of the P-value (adjusted for false-discovery rate, FDR) is plotted. The inset shows a magnification of the 'plasma membrane' term, which had a substantial increase in significance in the S7 fraction. (C) Relationship between enrichment slope and frequency of Ensembl-annotated transmembrane helicies. Each protein identified was assigned 1 if they were annotated as having a transmembrane helix, and 0 if not. All proteins were sorted according to enrichment slope, and an average was calculated every 50 proteins. (D) Stereocilia membrane protein enrichment. Displayed are bundle-enriched (>three-fold) proteins from *Wilmarth et al. (2014)* that were annotated as containing transmembrane domains, plus SLC34A2, SLC9A9, and SLC9A6. ND, not detected. (E) Clustering analysis of shotgun mass-spectrometry data using mclust. After data were log-transformed, they were normalized within a row, subtracting the row mean and dividing by the row standard deviation (thus the resulting mean was 0 and the standard deviation was 1). The best solution returned 25 clusters, which were ordered here by the enrichment slope (only the five fractions used in A were used to calculate the slope). PTPRQ, ATP2B2, and XIRP2 were all in the cluster with the steepest enrichment slope. (F–I) Targeted proteomics analysis of enrichment fractions. (F) Chromatograms of SAPLHILTDEDAPSSPPESLSVK peptide from PTPRQ detected in S1 (dark gray) and S7 (orange) fractions; inset shows magnified S1 chromatogram. Sum of 12 daughter ions. Arrows indicate where multiple MS2 spectra matched to PTPRQ were acquired. (G) Example MS2 spectrum from S7 sample; multiple peaks matched to predicted PTPRQ peptide fragment ions. (H) Targeted proteomics analysis of enrichment fractions. The mean of the total intensity area of 4–8 peptides per protein is indicated. (I) Same data as G, except area relative to S1 is indicated.

The following source data is available for figure 2:

**Source data 1.** Analysis of the shotgun proteomics experiments characterizing the protein composition of the D10 stereocilia-membrane purification from the chick inner ear.

**Source data 2.** Output of the mclust analysis of the D10 purification fractions.

for each fraction; we divided the intensities from peptides only found in each isoform by the sum of all isoform-specific peptides, generating the fractional contribution for each isoform. After making this adjustment, ATP2B2 was also one of the most enriched proteins (*Figure 2A*).

Gene ontology analysis (*Ashburner et al., 2000*; *Gene Ontology Consortium, 2015*) using DAVID (*Huang et al., 2009b, 2009a*) indicated that the ten most enriched GO terms in S7 relative to S1 referred to membranes, mitochondria, or plasma membrane (*Figure 2B*); the latter term was most highly enriched in the last enrichment fraction (p=3 $\times$ 10$^{-6}$). Similarly, 'membrane' is the most highly enriched (p=3 $\times$ 10$^{-37}$) keyword from the Protein Information Resource (*Barker et al., 2000*). In addition, the steeper the enrichment slope, the more likely a protein was annotated as containing a transmembrane helix (*Figure 2C*). Taken together, our analysis shows that the D10 preparation

**Table 2.** D10 purification. Protein assays and PTPRQ targeted proteomics assays were carried out on two preparations, each of ~1000 chick inner ears. Total PTPRQ PRM intensity was determined by multiplying the PTPRQ intensity from PRM targeted-proteomics experiments (per µg protein) by the total amount of protein (in µg) in each sample.

| Fraction | Description | Total protein (µg), ± range | PTPRQ PRM intensity, total | PTPRQ PRM intensity, fraction of PNS |
|---|---|---|---|---|
| S1 | Post nuclear supernatant | 109,000 ± 4,000 | 2.63 ± 0.40 × 10$^{10}$ | 100 ± 15% |
| S2 | 7000 rpm supernatant | 100,000 ± 4,000 | 0.35 ± 0.05 | 13 ± 3% |
| P2 | Dense membranes | 36,400 ± 700 | 2.56 ± 0.17 | 97 ± 16% |
| M3 | D10 bound | 2680 ± 90 | 0.34 ± 0.04 | 13 ± 2% |
| S6 | Cytoplasm | 29,200 ± 100 | 0.32 ± 0.07 | 12 ± 2% |
| P6 | Membranes | 984 ± 1 | 0.28 ± 0.01 | 11 ± 2% |
| S7 | RIPA-soluble | 1037 ± 0 | 1.13 ± 0.11 | 43 ± 8% |

substantially enriches membranes, especially stereocilia membranes, reducing protein complexity so that additional affinity purification steps can be used to identify protein complexes of stereocilia membranes.

The most abundant membrane proteins in S7 were either from the plasma membrane (e.g., ATP1A1, at riBAQ = $1.1 \times 10^{-2}$) or mitochondrial membranes (e.g., ATP5A1, at riBAQ = $1.1 \times 10^{-2}$). In comparison, ATP2B2, marking stereocilia membranes, was present at riBAQ = $0.7 \times 10^{-2}$. In frog stereocilia, ATP2B2 and ATP1A1 each have membrane densities of ~3000 $\mu m^{-2}$ (*Yamoah et al., 1998*; *Burnham and Stirling, 1984*); in rat liver mitochondria, ATP5A1 has a density of ~7500 $\mu m^{-2}$ (*Schwerzmann et al., 1986*). Taken together, these data suggest that stereocilia membranes account for ~5% of the solubilized membrane material in S7.

We used cluster analysis with mclust (*Fraley and Raftery, 2002*), a model-based hierarchical clustering algorithm that allows estimation of the best number of clusters for a dataset, to determine which proteins shared behavior across the stereocilia enrichment procedure. Using only proteins that were reproducibly detected across the preparation, mclust indicated that 25 clusters were appropriate for the normalized data (*Figure 2E*; *Figure 2—source data 2*). To determine which of these clusters were preferentially enriched in the D10 preparation, we applied a linear fit to the data from the sequential enrichment fractions (S1, P1, M3, P6, and S7). Cluster 8 had the greatest enrichment slope, and not surprisingly, it contained both PTPRQ and ATP2B2, as well as XIRP2 (*Figure 2E*).

We also used targeted proteomics to confirm that proteins of interest were enriched in the D10 preparation (*Figure 2F–I*). By directly examining the intensity associated with a known peptide, targeted proteomics usually offers substantially greater sensitivity and selectivity than shotgun proteomics with label-free quantitation (*Picotti and Aebersold, 2012*). We used PRM (*Gallien and Domon, 2015*; *Ronsein et al., 2015*) with an Orbitrap Fusion mass spectrometer to measure two to four peptides for each protein, monitoring at least two daughter ions per peptide. To confirm that the assays measured the peptide of interest, we spiked in heavy-labeled standard peptides; we also matched MS2 spectra collected during peaks to the protein of interest. Using this approach, we determined that PTPRQ was enriched between S1 and S7 by around 50-fold (*Figure 2*); the enrichment of ATP2B2 was less, perhaps because substantial amounts were located in hair-cell somas.

## Immunoaffinity purification of MYO7A complexes from chick stereocilia

We sought to identify tight membrane complexes containing MYO7A. MYO7A is known to interact with protein complexes located at stereocilia membranes, including those of tip links and ankle links, and is associated with membranes in other cells (*Soni et al., 2005*). While MYO7A is also present in hair cell somas, a rough estimate based on the amount in chick hair bundles (*Table 3*) indicates that purification would require an enrichment of approximately $10^{6}$-fold from dissected inner ears. The $10^{2}$-fold purification afforded by the stereocilia-membrane enrichment indicates that additional

**Table 3.** Identification of MYO7A-binding proteins in twist-off-purified hair bundles. The presented data, as well as technical aspects of the chick and mouse hair-bundle purification and shotgun mass spectrometry protein detection and quantitation, have been described previously (*Krey et al., 2015*; *Shin et al., 2013*; *Wilmarth et al., 2015*). BUN only, only found in hair bundle samples. UTR only, only found in whole utricle samples. ND, not detected.

| Protein | Chick molecules per stereocilium | Chick enrichment | Mouse (P23) molecules per stereocilium | Mouse (P23) enrichment |
|---|---|---|---|---|
| MYO7A | 796 | 1.6x | 229 | 28x |
| MYO6 | 9300 | 2.2x | 1305 | 2.6x |
| GIPC3 | 494 | 3.5x | 55 | 12x |
| MYO1C | 126 | 17x | ND | UTR only |
| MYO1H | 91 | 29x | 128 | 406x |
| ANKRD24 | 56 | 122x | 1 | BUN only |
| PDZD7 | 50 | 10x | ND | ND |
| SORBS1 | 16 | 2.0x | ND | UTR only |
| LMO7 | 10 | 3.4x | ND | ND |

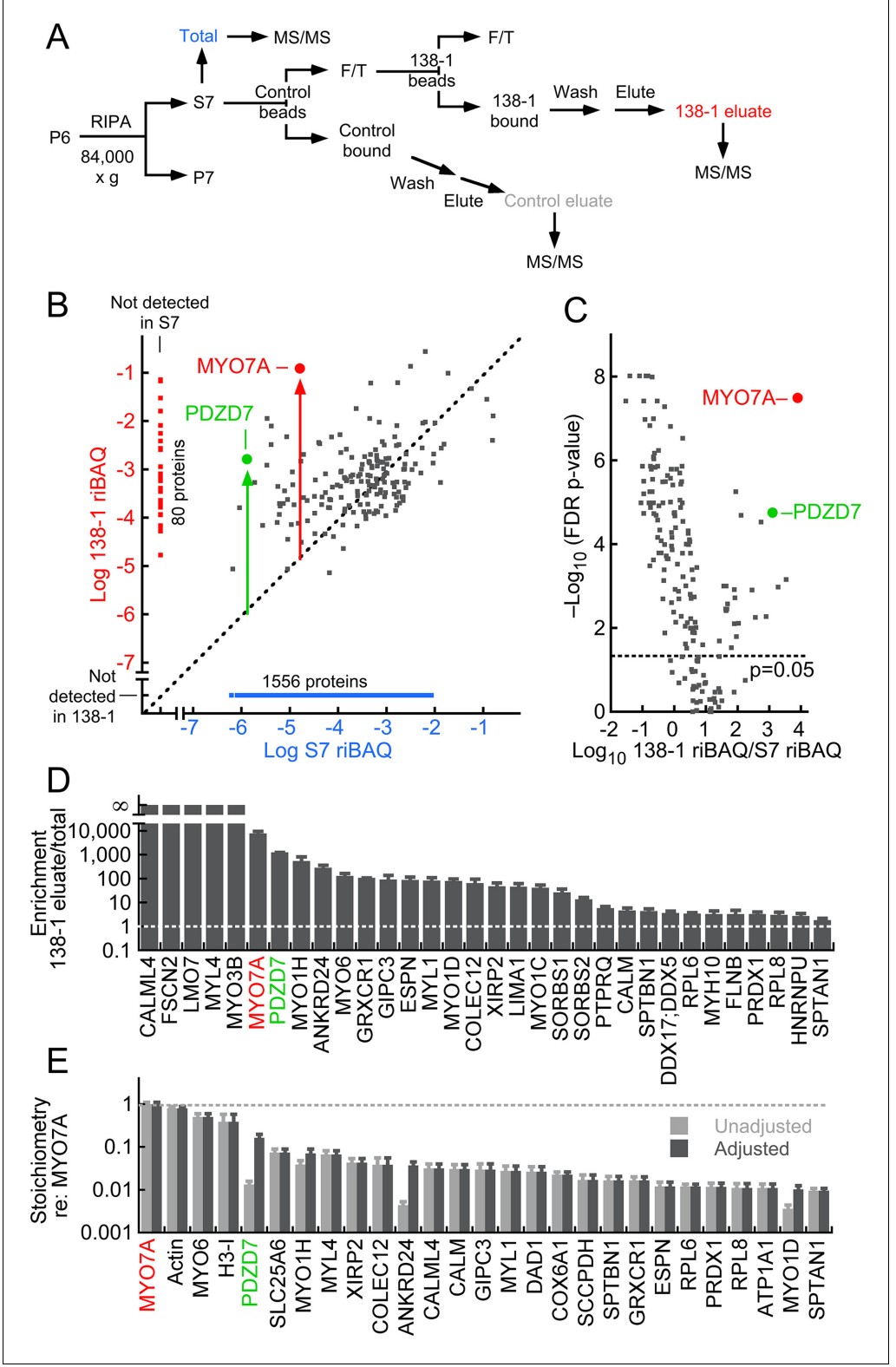

**Figure 3.** MYO7A immunoaffinity purification from D10 enriched membranes. (**A**) Flow chart describing immunoaffinity purification strategy using the 138-1 monoclonal antibody, which recognizes chicken MYO7A. Mass spectrometry was carried out on the total (S7; RIPA-soluble fraction), 138-1 eluate, and control eluate. F/T, unbound material; MS/MS, tandem mass spectrometry analysis. (**B**–**E**) Shotgun proteomics analysis of 138-1 immunoaffinity purification. (**B**) Relative protein levels of 1719 proteins detected in at least one run of total or 138-1

*Figure 3 continued on next page*

*Figure 3 continued*

eluate. Distance from the diagonal unity line (gray dashed) indicates enrichment by 138-1; note that MYO7A (red symbol) and PDZD7 (green) are highly enriched. (C) Volcano plot illustrating enrichment and statistical significance. The x-axis displays the log10 of each protein's enrichment by 138-1 immunoprecipitation relative to the S7 starting material, while the y-axis indicates the log10 value of the FDR (false discovery rate) adjusted p-value for that enrichment value. Dashed line indicates significance at p=0.05. (D) Top 32 proteins in 138-1 eluates by enrichment. Only proteins not detected in control runs and detected in at least four of six 138-1 runs are displayed. (E) Top 27 proteins in 138-1 eluates by stoichiometry relative to MYO7A. Only proteins detected in at least four of six 138-1 runs are displayed; proteins detected in control eluates (e.g., actin) were also included. Unadjusted stoichiometry: (138-1 riBAQ for protein of interest)/(138-1 riBAQ for MYO7A). Adjusted stoichiometry: (unadjusted stoichiometry) × (total riBAQ for MYO7A)/(total riBAQ for protein of interest). *Figure 3—figure supplement 1* displays an immunoblot analysis of the 138-1 immunoprecipitation, examining MYO7A and its Usher syndrome partners USH1C and CDH23.

The following source data and figure supplement are available for figure 3:

**Source data 1.** Analysis of the shotgun proteomics experiments characterizing the protein composition of the 138-1 anti-MYO7A immunoaffinity purification from the chick inner ear.
**Figure supplement 1.** Immunoprecipitation of upper tip-link density components from D10-enriched stereocilia membranes.

affinity purification steps are required to bring MYO7A complex abundance to the level where it can be definitively detected in shotgun mass-spectrometry experiments.

We therefore used immunoaffinity methods to further enrich MYO7A after purification of stereocilia membranes (*Figure 3A*); although the majority of MYO7A was in the insoluble fraction (P7) of the RIPA extraction of D10 membranes, a substantial amount was solubilized (S7). For the subsequent step, we used the 138-1 monoclonal antibody, which recognizes chicken MYO7A (*Soni et al., 2005*). We applied the S7 RIPA extract to control beads, constructed with purified mouse IgG, then applied the unbound material to 138-1 beads. Each aliquot of beads was washed thoroughly, then eluted with SDS; eluted proteins were then analyzed by shotgun and targeted mass spectrometry. Passing the extracts over control beads allowed us to identify proteins that nonspecifically bound to the immunoaffinity reagents, many of which were abundant; these proteins were then computationally subtracted from the proteins identified in the 138-1 eluates. We carried out three independent immunoprecipitation experiments, each with ~500 ear-equivalents; one to three technical replicates from each eluate were subjected to shotgun mass spectrometry, and the results from the six shotgun runs were analyzed together. All data are deposited at ProteomeXchange with the identifier PXD004221, and the analysis is tabulated in *Figure 3—source data 1*.

To characterize the immunoaffinity purification procedure, we first investigated whether previously-described MYO7A complexes were present in the 138-1 immunoprecipitates. Although a MYO7A-USH1C-USH1G-CDH23 complex has been inferred from cell-culture and complex reconstitution experiments (*Boëda et al., 2002*; *Wu et al., 2011*), no evidence has been presented to date for direct association of these proteins in the inner ear. When we immunoprecipitated MYO7A from stereocilia S7 detergent extracts (*Figure 3—figure supplement 1A*) and examined eluates using immunoblotting, we detected both a significant amount of USH1C (*Figure 3—figure supplement 1B*) and a small amount of CDH23 (*Figure 3—figure supplement 1C*) in the immunoprecipitates.

Because MYO7A and its interacting proteins should be highly enriched by the 138-1 antibody from the starting material, we analyzed immunoaffinity purification experiments using shotgun mass-spectrometry, comparing the relative abundance of each protein in the 138-1 eluate and in the stereocilia RIPA extract. This ratio—the immunoaffinity enrichment—will be high for MYO7A itself and for proteins that specifically coprecipitated with MYO7A. *Figure 3B* shows all proteins detected in the experiment, with those detected only in the 138-1 immunoprecipitate or in S7 segregated from those detected in both. For proteins detected in both fractions, the presence of a protein to the left and above the unity line (dashed) indicates that it is enriched; as expected, MYO7A itself was highly enriched (~$10^4$-fold). The statistical significance, adjusted for the false-discovery rate (FDR), is

indicated in a volcano plot (*Figure 3C*). MYO7A and PDZD7 were not only highly enriched, but the enrichment was highly significant statistically (p=$3 \times 10^{-8}$ and p=$2 \times 10^{-5}$).

We examined the proteins that were detected at least 4/6 times in the 138-1 runs and that were not detected in the control IgG eluates (*Figure 3D*). Remarkably, most of the most highly enriched proteins have been previously detected as key hair-bundle proteins (*Shin et al., 2013*); these include myosin motors MYO3B, MYO1H, and MYO6; actin-associated proteins ESPN, LMO7, FSCN2, and GRXCR1; and scaffolding proteins PDZD7 and GIPC3 (*Table 3*). Two other scaffolding proteins, SORBS1 and SORBS2, were also detected. Myosins apparently are not associated by binding to co-purifying actin filaments; actin was present at a low stoichiometry relative to MYO7A (<1 mol/mol), indicating that the concentration of filaments was very low. Moreover, treatment of immunoprecipitated material with 5 mM Mg•ATP did not release the other myosins from MYO7A complexes. We also noted the presence of multiple calmodulin-related proteins, including MYL4, CALML4, MYL1, and CALM itself; these molecules are candidate light chains for MYO7A or co-precipitating myosins. In these shotgun experiments, we did not detect USH1G or CDH23 in 138-1 samples; we did, however, detect USH1C in 138-1 eluate, albeit only in 2/6 runs.

PDZD7 was highly enriched along with MYO7A; both were present in the 138-1 immunoaffinity purification at relative concentrations >1000-fold higher than in S7 (*Figure 3B–D*). While the calculated stoichiometry of PDZD7 relative to MYO7A was low (~1%), PDZD7 was present in the stereocilia membrane extract at a far lower level than MYO7A. Indeed, a large fraction of the PDZD7 (~20%) was precipitated from the extract, indicating that many PDZD7 molcules were present in MYO7A complexes (*Figure 3E*). PDZD7 has been localized to the ankle links (*Grati et al., 2012*), where it is essential for localization of protein complexes containing ADGRV1, USH2A, and DFNB31 (*Chen et al., 2014*; *Zou et al., 2014*). Depending on the splice form, PDZD7 has two or three PDZ domains, and is a paralog of both USH1C and DFNB31 (*Ebermann et al., 2010*). An interaction of PDZD7 with MYO7A has not been reported previously, although the interactions of MYO7A with USH1C (*Boëda et al., 2002*) and DFNB31 (*Delprat et al., 2005*) suggest that the MYO7A-PDZD7 complex is plausible.

Containing five ankyrin domains, ANKRD24 is enriched in chick and mouse hair bundles (*Shin et al., 2013*; *Wilmarth et al., 2015*; *Krey et al., 2015*); little else is known about ANKRD24, although it is a hair-cell-specific protein in the chick ear (*Ku et al., 2014*). By shotgun mass spectrometry analysis, most of the ANKRD24 in the S7 extract was present in MYO7A complexes (*Figure 3E*).

## Targeted proteomics detection of MYO7A complexes in chick stereocilia

We used targeted proteomics to confirm that proteins identified in the shotgun analysis were indeed co-immunoprecipitated with MYO7A. We developed assays not only for several proteins detected in the shotgun experiments, but also for other proteins predicted to be in MYO7A complexes at tip links and ankle links. These experiments used heavy-labeled standard peptides to confirm retention times and daughter-ion patterns, and we also matched MS2 spectra to peaks to confirm the assays' veracity. Three biological replicates of 500 chick ears were used, with two technical replicates from each. Each peptide was analyzed in two or three of the biological replicates, yielding 4–6 total replicates for each. The analysis is tabulated in *Figure 4—source data 1*, and displayed in *Figure 4*.

For most proteins we examined, we detected strong peaks for peptides of interest in 138-1 immunoprecipitates but not in controls (*Figure 4*). For example, we detected daughter ions of the MYO7A peptide TTLTDSATTAK in 138-1 but not control samples (*Figure 4D*, left); this peak was identified definitively by matching an MS2 spectrum obtained during the peak to MYO7A in the database (*Figure 4D*, middle). This peptide was also detected in the extract used as the starting material (total; not shown). We examined four peptides, each of which displayed roughly equivalent levels of intensity in 138-1 immunoprecipitates as compared to totals (*Figure 4D*, right). Because the amount of material loaded in total samples (~2 ear-equivalents) was much less than loaded for the immunoprecipitation samples (~100 ear-equivalents), the y-axis reflects neither recovery nor enrichment but is useful for relative comparison of different proteins.

As in the shotgun proteomics experiments (*Figure 3*), MYO6 was readily detected in the eluates of 138-1 immunoprecipitates but not in controls (*Figure 4E*). Although a smaller fraction of MYO6 in the S7 extract was precipitated as compared to MYO7A (right panels), MYO6 was >10-fold more abundant in the extract than MYO7A, suggesting that MYO6 and MYO7A could be present in a

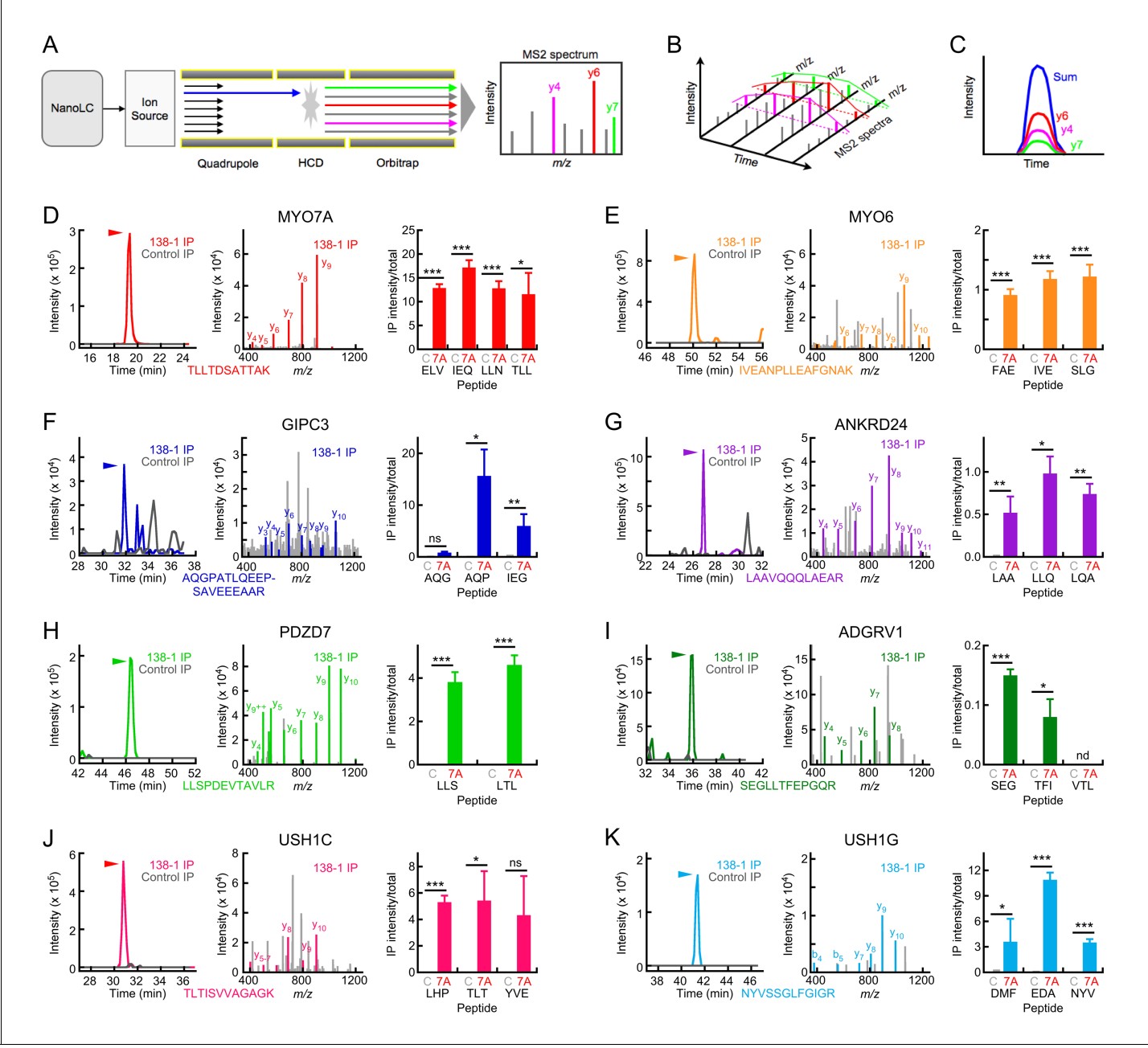

**Figure 4.** Targeted mass spectrometry analysis of MYO7A immunoaffinity purification from chick stereocilia. (**A–C**) Diagrams illustrating targeted analysis. (**A**) Peptides eluting from the nano-scale liquid chromatography (NanoLC) system are introduced into a Tribrid Fusion mass spectrometer by electrospray (Ion source). Monitored peptides (blue) are selected within the quadrupole mass analyzer, then are fragmented using higher-energy collisional dissociation (HCD). Fragment mass spectra (MS2) acquired from the Orbitrap analyzer are searched against a protein database to confirm the identity of the monitored peptide. (**B**) Robustly detected daughter ions (here y4, y6, and y7) can be monitored over time. (**C**) Each daughter ion is monitored over time; signal from all monitored daughter ions is summed and the time-summed intensity plot is integrated to determine the signal for the peptide of interest. (**D–K**) Targeted mass-spectrometry analysis of indicated proteins. Panels are: left, time-intensity plot summed over all daughter ions of indicated peptide (138-1 and control eluates); center, database-matched MS2 spectrum (138-1 eluate); right, integrated intensities for indicated peptides (mean ± SEM). (**D**) MYO7A. (**E**) MYO6. (**F**) GIPC3. (**G**) ANKRD24. (**H**) PDZD7. (**I**) ADGRV1. (**J**) USH1C. (**K**) USH1G. Key: **C**, control eluates; **7A**, 138-1 eluates. Arrowhead indicates elution position for heavy-labeled peptide and region where MS2 spectra match to the indicated protein. Statistical tests used two-tailed Student's t-test, with significance indicated in the figure as follows: *p<0.05; **p<0.01; ***p<0.001. Exact p-values and 95% confidence intervals are tabulated in *Figure 4—source data 1*.

The following source data is available for figure 4:

*Figure 4 continued on next page*

*Figure 4 continued*

**Source data 1.** Analysis of the targeted proteomics experiments characterizing the protein composition of the 138-1 anti-MYO7A immunoaffinity purification from the chick inner ear.

nearly equimolar complex. GIPC3 and ANKRD24 were also specifically precipitated using the 138-1 antibody (*Figure 4F,G*).

Finally, targeted proteomics allowed us to confirm that USH1G is in a complex with MYO7A (*Figure 4K*), as is USH1C (*Figure 4J*). Our targeted proteomics assay for CDH23 was unfortunately not sensitive enough to detect it in the 138-1 eluates. Our combined immunoblotting and targeted proteomics data thus indicates that a MYO7A complex in stereocilia with USH1C, USH1G, and—probably—CDH23 is plausible.

Other proteins known to be located in ankle links also co-precipitated with MYO7A. While we did detect ADGRV1 and DFNB31 in 138-1 immunoprecipitates by shotgun proteomics, their inconsistent appearance (identified in only two of six mass-spectrometry runs each) prevented us from definitively concluding that they were in a complex with MYO7A. To provide stronger evidence that MYO7A forms a protein complex with PDZD7 and ADGRV1, we used targeted proteomics to determine how much of each protein was precipitated by 138-1 anti-MYO7A or control immunoaffinity beads. PDZD7 was present at significantly greater levels in 138-1 precipitations than in control precipitations (*Figure 4H*), confirming its specific precipitation with MYO7A. By contrast, only one peptide for ADGRV1 showed significant enrichment in 138-1 immunoprecipitates compared to controls, and the fraction precipitated was very small (*Figure 4I*). We suggest that PDZD7:MYO7A complexes are more abundant than ones containing ADGRV1, or that ADGRV1 binds less tightly than PDZD7.

## Interaction of mouse MYO7A and PDZD7 in tissue-culture cells

To confirm that PDZD7 interacts directly with MYO7A and extend the results to mammals, we co-expressed mouse or human proteins in HEK293 cells and immunoprecipitated GFP-MYO7A with the 138-1 monoclonal antibody. When detected using an anti-PDZD7 antibody, PDZD7 fused to mCherry was specifically immunoprecipitated by 138-1 along with GFP-MYO7A (*Figure 5A*). The MYO7A-PDZD7 stoichiometry was high, as the majority of the PDZD7 was immunoprecipitated. ANKRD24 also specifically co-immunoprecipitated with MYO7A (*Figure 5B*), although a much lower fraction of the total ANKRD24 was recovered. By contrast, PDZD7 tagged with Strep-tag II did not co-immunoprecipitate with ANKRD24-GFP, supporting the specificity of the interactions of PDZD7 and ANKRD24 with MYO7A (*Figure 5C*).

To determine whether any other HEK proteins were utilized in making the MYO7A-PDZD7 complex, we analyzed 138-1 immunoprecipitates by shotgun mass spectrometry. All data are deposited at ProteomeXchange with the identifier PXD004266. All analyses are presented in *Figure 5—source data 1*. As expected, neither MYO7A nor PDZD7 were detected in untransfected HEK cells (*Figure 5D*); when PDZD7 alone was expressed, it was present at a relatively high total fraction in the HEK cell extract but was not immunoprecipitated by 138-1. By contrast, when MYO7A was co-expressed, PDZD7 was present in the 138-1 immunoprecipitate at a level ~0.4x that of MYO7A, confirming a near-stoichiometric interaction. No other proteins were co-precipitated at similar abundance, with the exception of one abundant peptide for the protein FER1L1, which was also artifactually precipitated by 138-1 in the untransfected controls.

To visualize the PDZD7-MYO7A interaction in tissue-culture cells, we used the nanoscale pull-down method (*Bird et al., 2016*), which relies on a fusion between a binding partner and a MYO10 'heavy meromyosin-like' construct, which contains the motor, light-chain-binding region, and single-alpha-helix domain. This fusion protein—and any co-associating proteins—specifically concentrate at the tips of COS7 filopodia, where they are easily visualized (*Berg and Cheney, 2002*; *Kerber and Cheney, 2011*). The human MYO7A tail (residues 967–2175, including the two tandem MyTH4-FERM domains and the SH3 domain) was fused to a construct containing GFP and bovine MYO10 heavy meromyosin (MYO10$_{HMM}$; motor and light-chain binding domains). Co-expression of the MYO10$_{HMM}$-MYO7A$_{tail}$ fusion with mCherry-USH1G led to robust targeting to filopodia tips (*Figure 6A,B*). As seen previously, mCherry-USH1C formed intracellular actin cables; co-expressed MYO10$_{HMM}$-MYO7A$_{tail}$ targeted there too (*Figure 6B*).

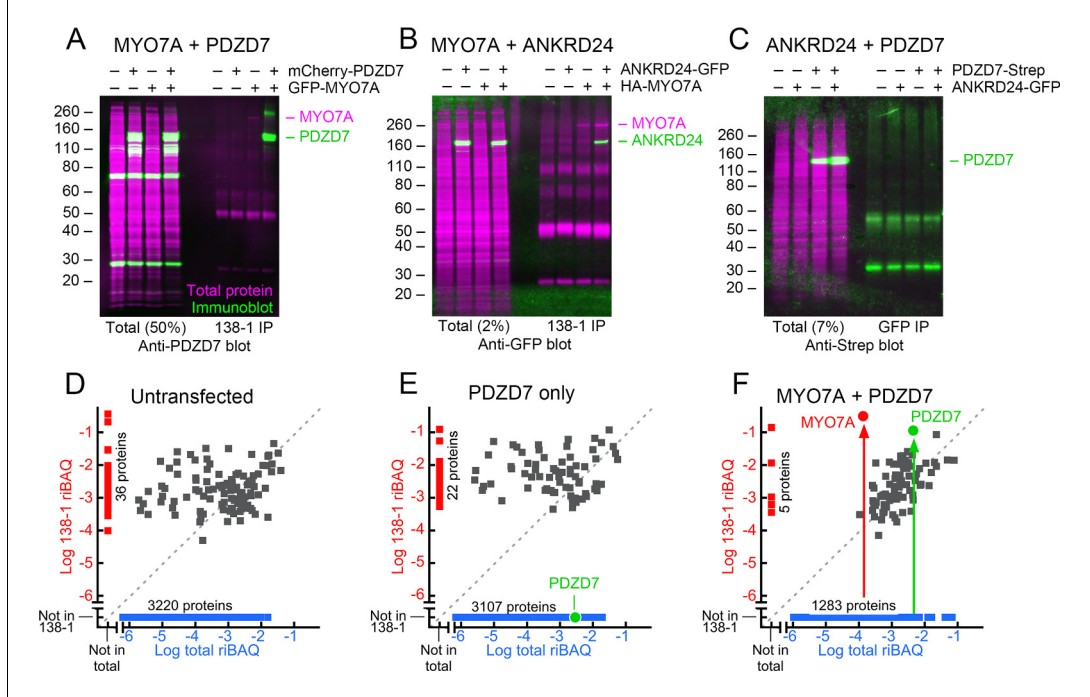

**Figure 5.** Interaction of mouse MYO7A and partners in HEK293 cells. PDZD7, ANKRD24, and GFP-MYO7A proteins are mouse; HA-MYO7A is human. (A–C) Immunoprecipitation and protein immunoblotting. Each panel has the immunoblot detection (green) superimposed on the ink stain for total protein (magenta). Note that MYO7A, immunoprecipitated in each case, is usually visible on the ink stain. Molecular mass markers (in kD) are indicated on the left. Left side, starting material for immunoprecipitation (total), with the fraction loaded relative to the immunoprecipitate indicated. Right side, immunoprecipitates. (A) PDZD7 and MYO7A. Immunoprecipitation with 138-1 antibody; immunoblot with anti-PDZD7. (B) ANKRD24 and MYO7A. Immunoprecipitation with 138-1 antibody; immunoblot with anti-GFP. (C) PDZD7 and ANKRD24. Immunoprecipitation with anti-GFP antibody; detection with anti-Strep. (D–F) Immunoprecipitation and mass spectrometry. (D) Untransfected HEK293 cells. (E) HEK293 cells transfected with *mCherry-Pdzd7* plasmid alone. No PDZD7 was immunoprecipitated with the 138-1 antibody. (F) HEK293 cells co-transfected with *Gfp-Myo7a* and *mCherry-Pdzd7* plasmids. MYO7A and PDZD7 were highly enriched in the 138-1 immunoprecipitates. Note that no PDZD7 was detected in HEK293 cells transfected with *Gfp-Myo7a* alone.

The following source data is available for figure 5:

**Source data 1.** Analysis of the shotgun proteomics experiments characterizing the protein composition of the 138-1 anti-MYO7A immunoaffinity purification from transfected HEK cells.

mCherry-PDZD7 targeted to filopodia tips, but only if MYO10$_{HMM}$-MYO7A$_{tail}$ was expressed (*Figure 6D*); with a few exceptions, no mCherry-PDZD7 was clustered with the control construct MYO10$_{HMM}$ (*Figure 6C*). By contrast, only minimal amounts of ANKRD24-Strep targeted to filopodia, confirming its interaction with the MYO7A tail was weak, nonexistent, or masked in tissue-culture cells (*Figure 6E*).

## Normal hair-bundle morphology requires MYO7A

To determine whether PDZD7 depends on MYO7A for localization in hair cells, we compared wild-type and *Myo7a8J* mice, which are functionally null (*Zheng et al., 2012*). High-resolution confocal imaging revealed additional features of hair bundles in *Myo7a^8J* mouse utricles (*Figure 7A–G*). As previously reported, vestibular stereocilia of 8J/8J homozygotes were not arranged in a coherent bundle but instead projected randomly; moreover, stereocilia heights in *Myo7a^8J* mice were irregular, with many stereocilia being longer than the longest stereocilia in wild-type mice (*Figure 7B,F, M*). Not only were stereocilia not arranged in a staircase manner, but unlike in wild-type bundles (*Figure 7E*), they were often arranged radially around the cytoplasmic channel where the kinocilium inserts, the fonticulus (*Figure 7G*, asterisk). The fonticulus was usually found in the center of cuticular plate, indicating a loss of subcellular planar polarity (*Deans, 2013*).

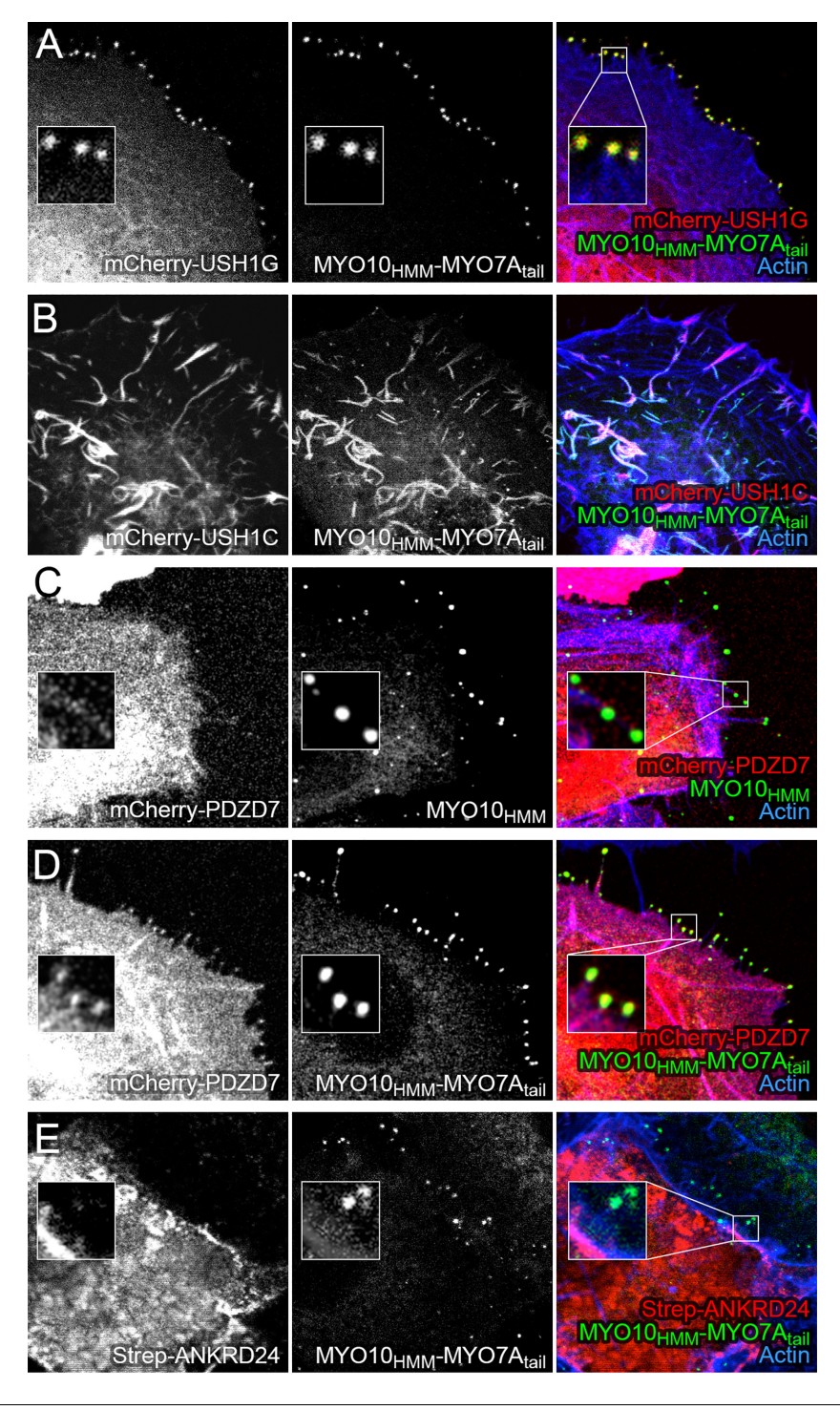

**Figure 6.** Co-localization of PDZD7 with MYO10-MYO7A fusion. COS7 cells were transfected with the indicated constructs and stained with phalloidin to detect actin. MYO10$_{HMM}$ and MYO10$_{HMM}$-MYO7A$_{tail}$ constructs (derived from bovine MYO10 and human MYO7A) were fused to GFP, which was imaged directly, as was mCherry fused to the indicated molecules. (**A**) mCherry-USH1G robustly localizes to filopodia tips when expressed with MYO10$_{HMM}$-MYO7A$_{tail}$. (**B**) mCherry-USH1C generates actin cables, to which MYO10$_{HMM}$-MYO7A$_{tail}$ targets. (**C**) mCherry-PDZD7 does not concentrate at filopodia tips when the control MYO10$_{HMM}$ construct is expressed. Occasional exceptions are seen (yellow asterisk). (**D**) mCherry-PDZD7 does concentrate at filopodia tips when the MYO10$_{HMM}$-

*Figure 6 continued on next page*

*Figure 6 continued*

MYO7A$_{tail}$ construct is expressed. (E) Very little Strep-ANKRD24 co-localizes with MYO10$_{HMM}$-MYO7A$_{tail}$ at filopodia tips. Main panels are 40 × 40 µm; inset panels are 4 × 4 µm.

Depending on the report, MYO7A localization in mouse vestibular hair bundles has been reported to be at stereocilia bases, at tip links, and throughout the bundle (*Hasson et al., 1997*; *Boëda et al., 2002*; *Senften et al., 2006*; *Grati and Kachar, 2011*). To localize MYO7A, we used an antibody that did not label 8J/8J homozygous bundles (*Figure 7B*); this antibody produced punctate labeling that was often stronger near stereocilia tips (*Figure 7A*). To allow imaging of utricle stereocilia by superresolution microscopy with structured-illumination (*Gustafsson, 2000*), we isolated stereocilia by absorption to poly-L-lysine-coated coverslips (*Neugebauer and Thurm, 1984*, *Hasson et al., 1997*); these experiments showed that MYO7A bound along the periphery of the actin core, juxtaposed with the membrane (*Figure 7C*).

As previously reported for cochlea (*Michalski et al., 2007*), the localization of ADVGRV1 to the ankle link region in utricle hair bundles depended on functional MYO7A (*Figure 7H–I*). By contrast, localization of ATP2B2 was unaffected in 8J/8J homozygous utricle bundles (*Figure 7J–K*).

To visualize MYO7A in cochlear hair cells, we used structured-illumination microscopy. MYO7A was punctate, with weak labeling in hair bundle, and stronger labeling in the cell soma (*Figure 7L*). As previously reported (*Self et al., 1998*), stereocilia of 8J/8J homozygotes were highly disorganized and had irregular lengths; no MYO7A immunoreactivity was detected (*Figure 7M*). By contrast, ATP2B2 localization in inner and outer hair cells was relatively normal, despite the bundle disarray; in both heterozygotes and homozygotes, labeling was much stronger in outer hair cells than in inner hair cells (*Dumont et al., 2001*; *Beurg et al., 2010*).

## PDZD7 localization at ankle link region in mouse inner ear

We sought to confirm the localization of PDZD7 at ankle links and to determine whether its location depended on MYO7A. To best visualize ankle links in a z-stack of x-y confocal images, we maximum-projected x-z slices to a depth of 5–10 µm, allowing superposition of multiple hair bundles in a profile perspective. As previously shown (*Grati et al., 2012*; *Zou et al., 2014*), PDZD7 localized to ankle links of heterozygous P8 *Myo7a*$^{+/8J}$ utricle hair cells (*Figure 8A*). Some cell bodies of hair cells had large amounts of PDZD7 immunoreactivity (*Figure 8A,F*), and PDZD7 was located in the pericuticular necklace region and the funiculus (*Figure 8B*). Both regions have been implicated in vesicular trafficking.

We also imaged PDZD7 using structured-illumination microscopy of isolated stereocilia; both intact bundles (*Figure 8C*) and dispersed stereocilia (*Figure 8D*) were obtained. When we labeled isolated stereocilia with antibodies against PDZD7, some labeling was evident near basal tapers of stereocilia (*Figure 8C*). When we quantified PDZD7 and MYO7A labeling (*Figure 8D–E*), we noted consistent PDZD7 labeling at stereocilia tapers, despite significant background labeling; MYO7A labeling was more dense in distal regions of the stereocilia. There was some overlap between PDZD7 and MYO7A labeling at stereocilia tapers, however (*Figure 8D*, right panel, inset).

To determine whether PDZD7 and MYO7A co-localize in intact hair bundles, we used confocal imaging with deconvolution processing. While MYO7A and PDZD7 generally did not overlap, some punctae within the stereocilia were labeled clearly with both antibodies (*Figure 8F*).

## Transfected mCherry-PDZD7 also targets to ankle link region

To confirm the localization of PDZD7 at the stereocilia ankle link region, we transfected hair cells with mCherry-PDZD7 and investigated its location. We first used in utero electroporation of E12 mouse otocysts (*Brigande et al., 2009*), then analyzed utricles at P2. In two transfected utricles, we identified >20 transfected hair cells; the mCherry signal was low in all, but in cells with the highest expression of mCherry-PDZD7, the signal was clearly discernable at the base of the hair bundle (*Figure 8G*).

We also used injectoporation (*Xiong et al., 2012*; *Zhao et al., 2014*; *Xiong et al., 2014*) to deliver the mCherry-PDZD7 plasmid to cochlear hair cells. Many cells were transfected, with a wide

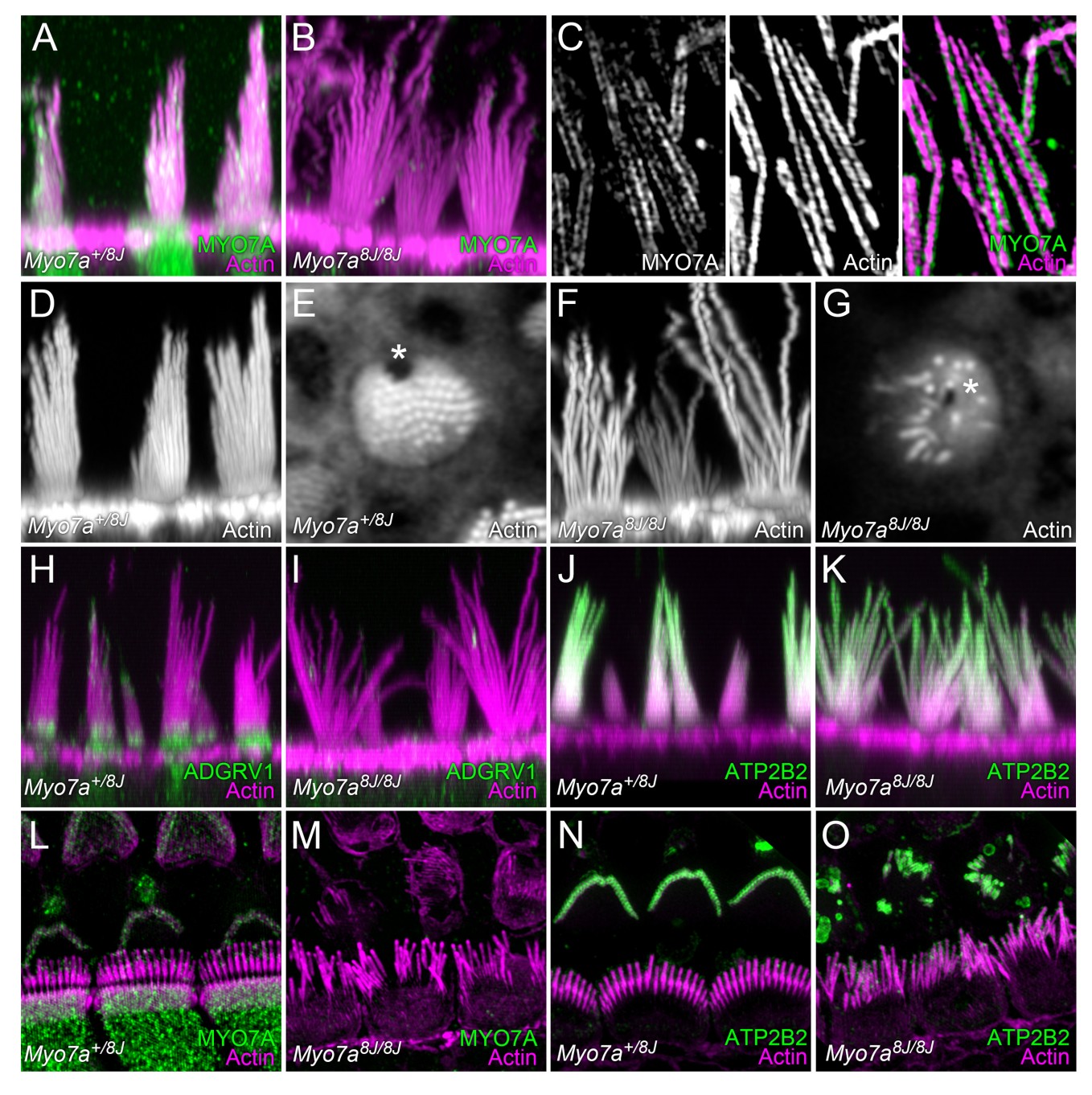

**Figure 7.** MYO7A localization in P8 mouse hair cells. (**A**) MYO7A immunoreactivity in +/8J heterozygote utricle. (**B**) MYO7A immunoreactivity in a 8J/8J homozygote utricle. Antibody signal is gone. (**C**) MYO7A immunoreactivity in utricle stereocilia isolated on poly-lysine-coated glass; structured-illumination microscopy. Note punctate labeling outside of the actin core. (**D**) Phalloidin staining of +/8J heterozygote utricle. (**E**) Cross-section of +/8J heterozygote utricle immediately above the cuticular plate. The cytoplasmic channel where the kinocilium inserts, the fonticulus, is located asymmetrically (*). (**F**) Phalloidin staining of 8J/8J homozygote utricle. Many stereocilia are abnormally long; bundle cohesion is absent. (**G**) Cross-section of 8J/8J homozygote utricle immediately above the cuticular plate. Note the fonticulus is centrally located (*). (**H**) ADGRV1 immunoreactivity in +/8J heterozygote utricle; note band at ankle links region. (**I**) ADGRV1 immunoreactivity in a 8J/8J homozygote utricle. Antibody signal is gone. (**J**) ATP2B2 immunoreactivity in +/8J heterozygote utricle. (**K**) ATP2B2 immunoreactivity in a 8J/8J homozygote utricle. Antibody signal is unchanged. (**L**) MYO7A immunoreactivity in +/8J heterozygote cochlea; structured-illumination image. MYO7A labeling is punctate; it is sparse in the hair bundle, and strong in the cytoplasm. Inner hair cell stereocilia are bent backwards against the coverslip, causing them to overlap the outer hair cell area. (**M**) MYO7A immunoreactivity in a 8J/8J homozygote cochlea. Antibody signal is absent. (**N**) ATP2B2 immunoreactivity in +/8J heterozygote cochlea. Note weak staining in inner hair cells, but extremely strong staining in outer hair cells. (**O**) ATP2B2 immunoreactivity in a 8J/8J homozygote cochlea.

*Figure 7 continued on next page*

*Figure 7 continued*

Although the morphology of bundles from inner and outer hair cells has changed dramatically, ATP2B2 is still targeted to each in approximately the same density. Scales: panels A, B, D, and F are 20 × 20 µm; panels in C are 6.7 × 10 µm; panels E and G are 10 × 10 µm; panels H–K are 25 × 25 µm; panels L–O are 20 × 20 µm.

range of expression levels. Cells with the highest levels of mCherry signal had very disorganized hair bundles (*Figure 8—figure supplement 1*), a phenotype we rarely see in cells injectoporated with control plasmids (*Xiong et al., 2012*; *Zhao et al., 2014*; *Xiong et al., 2014*). By contrast, cells with the lowest levels of signal had mCherry-PDZD7 located specifically in hair bundles, largely at stereocilia bases (*Figure 8H–I*). We noted that many cells with intermediate levels of mCherry-PDZD7 had some signal located specifically at stereocilia tips (*Figure 8—figure supplement 1*).

## PDZD7 depends on MYO7A for localization at ankle link region

PDZD7 depended on MYO7A for localization to ankle links (*Figure 8J–O*). While localization was normal in heterozygous *Myo7a^{+/8J}* utricles (*Figure 8J–K*), when we examined 8J/8J homozygous *Myo7a^{8J/8J}* utricles at P8, we noted that PDZD7 was absent from the ankle-link region, although it remained present in the apical region of hair cell somas (*Figure 8L–M*). Strong PDZD7 appeared in the cytoplasmic channel where the kinocilium inserted; in some cases the labeling extended up into the hair bundle region, but was distinct from ankle-link labeling (*Figure 8L–M*).

In the cochlea, PDZD7 labeling was prominent in the ankle-link region of bundles of heterozygous cochlear inner hair cells visualized using structured-illumination microscopy (*Figure 8N*). By contrast, PDZD7 labeling was absent from homozygous *Myo7a^{8J/8J}* bundles (*Figure 8O*).

## Discussion

Biochemical purification of protein complexes allows determination of the composition of those complexes that may otherwise be missed with other methods, such as genetics. Unfortunately, each inner ear has only a few thousand hair cells, which constitute <1% of the total cell mass of the ear, and hair bundles make up only ~1% of the protein in a hair cell (*Shin et al., 2013*; *Krey et al., 2015*). Moreover, many of the most interesting proteins of the bundle, like those involved in mechanotransduction, are present at very low copy numbers per stereocilium; for example, there are only a few hundred MYO7A molecules for the 400,000 actin molecules in an average stereocilium (*Table 3*). Thus, while biochemical characterization of protein complexes is essential for unraveling bundle function, the challenges of low absolute and relative protein abundance have hitherto prevented large-scale complex isolation from stereocilia.

By scaling up to hundreds to thousands of chicken ears per preparation and by exploiting monoclonal antibodies specific for stereocilia proteins, we were able to use biochemical methods to characterize protein complexes from stereocilia membranes of hair cells. We demonstrated the utility of the D10 enrichment method by examining protein complexes that include MYO7A. While our strategy does not yield purified MYO7A complexes, as they are too scarce in the stereocilia-membrane extract, we identified candidates for complex members based on each protein's stoichiometry relative to MYO7A, immunoaffinity enrichment, and statistical significance of that enrichment.

Several potential roles have been ascribed to MYO7A, but it has not been clear how this motor protein can carry out disparate functions like positioning ankle links and controlling transduction. Our results show that MYO7A interacts with several scaffolding proteins, which suggests that different MYO7A complexes may simultaneously mediate several functions. We have focused here on PDZD7, which is essential for the formation of the ankle-link protein complex (*Ebermann et al., 2010*; *Grati et al., 2012*; *Zou et al., 2014*; *Chen et al., 2014*) and requires MYO7A for localization to ankle links. We also showed that MYO7A is present in inner ear tissue in the predicted complexes with USH1C, USH1G, and CDH23. Our results thus demonstrate that it is indeed possible to characterize mechanotransduction using biochemical methods, which complement genetics, physiology, and microscopy methods.

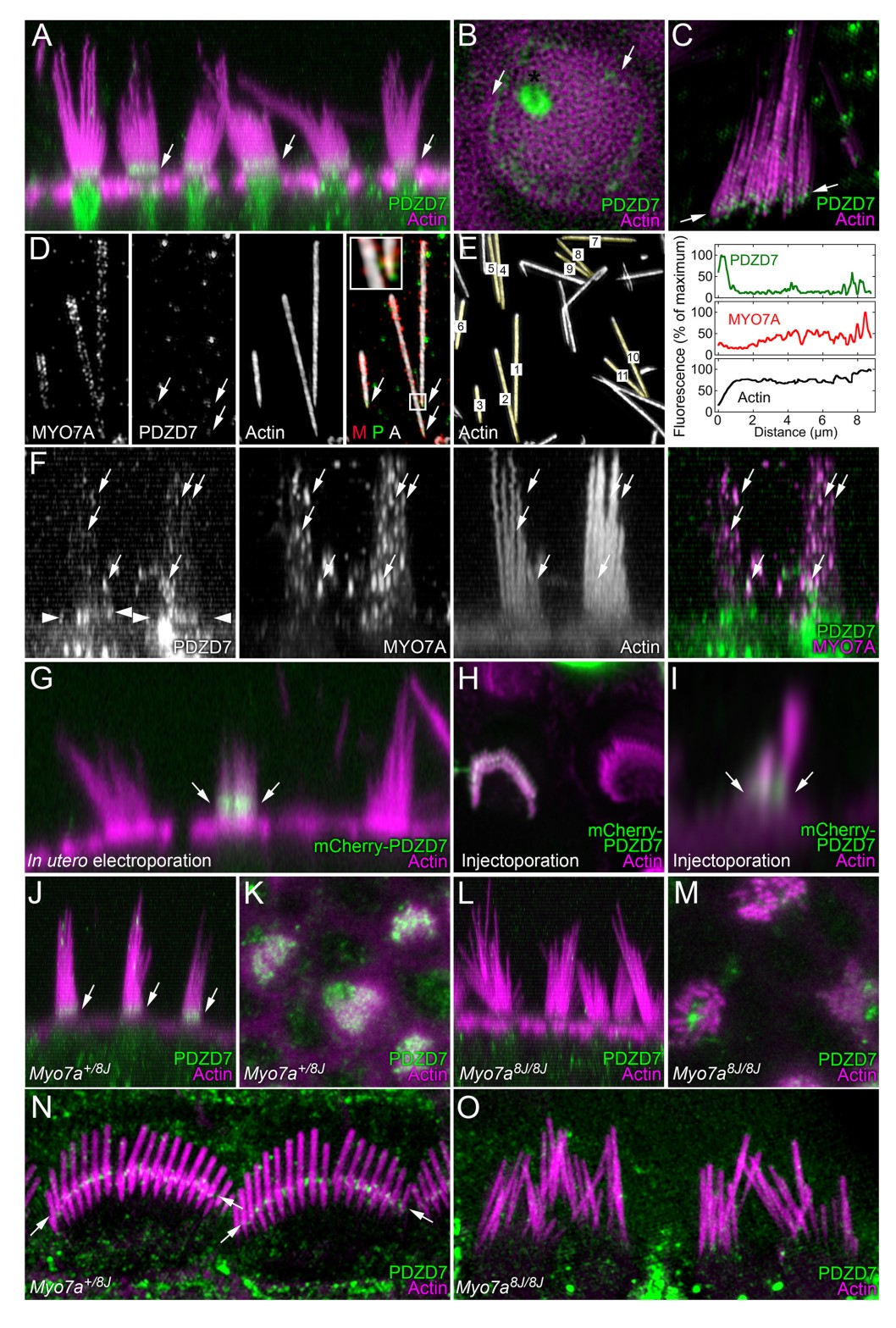

**Figure 8.** PDZD7 localization at ankle links depends on MYO7A in P8 mice. (**A**) PDZD7 immunoreactivity in wild-type utricle. Note PDZD7 in band near the stereocilia ankles (arrows), consistent with previous localization at ankle links. (**B**) Deconvolution analysis of utricle hair bundle at the level of the cuticular plate. PDZD7 immunoreactivity appears as a ring around the cytoplasmic channel in the cuticular plate. PDZD7 is also present in the pericuticular necklace surrounding the cuticular plate (arrows); image gamma was adjusted to 1.5 to allow visualization of

*Figure 8 continued on next page*

*Figure 8 continued*

periticular labeling without saturating cytoplasmic-channel labeling. (C) PDZD7 immunoreactivity in utricle hair bundle isolated on poly-lysine-coated glass; structured-illumination microscopy (SIM). Immunoreactivity near the stereocilia tapers is apparent (arrows). (D) Co-labeling of MYO7A and PDZD7 in utricle stereocilia isolated on glass using SIM. PDZD7 labeling above tapers is indicated (arrows). Insert, magnification of box in right-hand merged image. M, MYO7A; P, PDZD7; A, actin. Note that MYO7A and PDZD7 punctae occasionally overlap (yellow). (E) Quantitation of PDZD7 and MYO7A along utricle stereocilia length. Left panel, quantified stereocilia are indicated. Note that the three stereocilia from panel D are included. Right panel, average fluorescence signals from each channel from stereocilia aligned at the taper. Note the decreased stereocilia diameter near 0 μm, which corresponds to the peak of PDZD7 labeling (at ~0.5 μm). (F) MYO7A and PDZD7 punctae partially overlap in utricle stereocilia shafts. Individual channels and the merge of the PDZD and MYO7A channels are shown. Arrows indicate punctate that are present both in PDZD7 and MYO7A channels; arrowheads indicate PDZD7 at ankle-links region. (G) Single utricle hair cell after in utero electroporation with mCherry-PDZD7 at E12 and analysis at P2. mCherry-PDZD7 is located in the ankle-link region (arrows). (H) Single cochlear outer hair cell labeled after injectoporation with mCherry-PDZD7 at P3 and imaged after 2 days in culture. Labeling is associated with stereocilia base, although the top view obscures that localization. Airyscan processing. Additional transfected cells are displayed in *Figure 8—figure supplement 1*. (I) Another mCherry-PDZD7 cell, viewed in profile using x-z reslice. Note signal is concentrated near stereocilia bases (arrows). Airyscan processing. (J) PDZD7 immunoreactivity in utricle hair cells of *Myo7a*$^{+/8J}$ heterozygote mice. Labeling at ankle-link region is clear (arrows). (K) PDZD7 at the ankle region of heterozygote utricle. (L) PDZD7 in utricle hair cells of *Myo7a*$^{8J/8J}$ null mice. While PDZD7 is still in the cytoplasm, it is no longer also located near ankle links. Labeling in the cytoplasmic channel and kinocilium base remains. (M) PDZD7 at ankle region in homozygous utricles. Little or no PDZD7 is detected at ankle links. (N) PDZD7 in cochlea hair cells of *Myo7a*$^{+/8J}$ heterozygote mice using SIM. Labeling at ankle-link region is clear (arrows). (O) PDZD7 in cochlea hair cells of *Myo7a*$^{8J/8J}$ null mice using SIM. There is no PDZD7 immunoreactivity in bundles. Scales: panel A is 20 × 40.4 μm; panel B is 7.5 × 7.5 μm; panel C is 10 × 10 μm; panels in D are 6.4 × 13 μm; panels in F are 17.5 × 17.5 μm; panel G is 20 × 40.4 μm; panel H is 12 × 12 μm; panel I is 5 × 5 μm; panels J and L are 30 × 30 μm; panels K and M are 15 × 15 μm; panels N and O are 10 × 20.2 μm.

The following figure supplement is available for figure 8:

**Figure supplement 1.** Localization of mCherry-PDZD7 at tips of stereocilia.

## Protein complex purification from stereocilia membranes

Our original hair-bundle isolation method, which uses agarose to embed and excise the bundles, yields stereocilia that are >90% pure (*Gillespie and Hudspeth, 1991*; *Shin et al., 2013*; *Krey et al., 2015*). Considerable fine dissection is required, however, to prepare the inner-ear organs and clean up the agarose-embedded hair bundles; these steps make the daily throughput of bundles relatively low. In addition, the agarose matrix that extricates bundles complicates subsequent steps, such as detergent extraction of membrane proteins. While protein complexes can be detected by immuno-precipitation using the twist-off method (*Yamoah et al., 1998*; *Hill et al., 2006b*), the D10 purification strategy is far better for obtaining large amounts of stereocilia-membrane material that is suitable for complex extraction and purification (*Table 1*).

The D10 preparation has significant contamination, however; we estimate that stereocilia membranes account for only ~5% of the RIPA-solubilized material. Other membranes are present in part because contaminating intracellular organelles are disrupted by a freeze-thaw cycle, preventing their precise separation from stereocilia based on distinct, uniform sedimentation properties (*Cox and Emili, 2006*). Many of the most important stereocilia proteins are present at vanishingly small levels, however, so for stereocilia membrane enrichment, we chose to maximize total recovery of stereocilia membrane protein rather than purity. Immunoaffinity purification, especially with high-affinity mono-clonal antibodies, allows tremendous enrichment, even if multiple affinity enrichment steps are ultimately needed for some stereocilia protein complexes.

We stabilized protein complexes of the stereocilia using chemical crosslinking with membrane-impermeant crosslinker DTSSP, which should enhance our ability to detect low-abundance membrane protein interactions (*Vasilescu et al., 2004*; *Gokhale et al., 2012*; *Kim et al., 2012*; *Corgiat et al., 2014*). By using DTSSP, we were able to use stringent extraction and washing

conditions, minimizing the dissociation of protein complexes that included extracellular domains. Because the disulfide bond in DTSSP can be split by treating samples with a strong reducing agent prior to immunoblotting or mass spectrometry, we were able to examine proteins associated with MYO7A.

Were all proteins precipitated in our experiments with the 138-1 antibody in specific complexes with MYO7A? Probably not. While DTSSP should not penetrate into cells and hence artifactually crosslink nearby proteins, it is possible that trace amounts of large cytoskeletal complexes are purified with the immunoaffinity procedure. Subsequent validation of interactions detected, for example showing co-immunoprecipitation, protein-protein binding, or mislocalization in mutant hair cells, is thus of high importance.

## Association of MYO7A with PDZD7 and other ankle-link components

Our data show that MYO7A interacts with PDZD7, a key ankle-link component. Originally identified as a modifier gene for Usher syndrome type 2 genes, *PDZD7* has more recently been identified as a nonsyndromic deafness gene itself (*Booth et al., 2015*; *Vona et al., 2016*). Genetics experiments showed that positioning of ADGRV1 at ankle links depends critically on PDZD7 (*Zou et al., 2014*); consistent with those results, PDZD7 binds directly to ADGRV1 (*Chen et al., 2014*). Our experiments suggest how this positioning might occur; PDZD7 could bind to MYO7A, and then MYO7A could transport the PDZD7-ADGRV1 complex to its final location. While present at lower levels than PDZD7, we did detect ADGRV1 in co-immunoprecipitates with MYO7A by both shotgun and targeted mass spectrometry; our experiments do not reveal, however, whether all three are simultaneously in the same protein complex.

PDZD7 requires functional MYO7A for localization to ankle links. MYO7A is also required for localization of ADGRV1; while MYO7A might bind directly to ADGRV1, the necessity of PDZD7 for the formation of the ankle-link complex suggests that MYO7A localizes both PDZD7 and ADGRV1 at ankle links. Association of MYO7A with ankle-link proteins explains the early observation that MYO7A concentrates at the ankle-link region of vestibular stereocilia of mature frogs (*Hasson et al., 1997*) and cochlea stereocilia of immature mice (*Senften et al., 2006*; *Lĕfevre et al., 2008*). Positioning at this location by MYO7A is surprising, however, as the ankle links are very close to the pointed ends of the filaments located at the periphery of the stereocilia actin core. We suggest three possible explanations. First, because MYO7A moves from pointed to barbed ends of actin filaments (*Inoue and Ikebe, 2003*), MYO7A may move PDZD7 and ADGRV1 towards stereocilia tips but stall if the N-terminal, extracellular end of ADGRV1 was anchored on the adjacent stereocilium. In this case, MYO7A would put ankle links under tension, which could also explain the frequent oblique positioning of the links (*Goodyear and Richardson, 1992*, *1999*). Second, MYO7A could hand off PDZD7 to MYO6 anywhere in the stereocilium; because MYO6 moves towards pointed ends of actin filaments, it would properly localize the ankle-link proteins at stereocilia bases. Indeed, MYO6 is thought to move PTPRQ to the taper region (*Sakaguchi et al., 2008*). Finally, MYO7A and MYO6 may be in a multimolecular, bidirectional motor complex; like with other bidirectional motor complexes (*Hancock, 2014*), reciprocal regulation of the two motors could steer the complex to the top or bottom of the stereocilia. Indeed, MYO6 was the most abundant protein co-precipitated with MYO7A, and we are presently investigating the specificity of this interaction. Regardless of the mechanism, the data here show that MYO7A binds PDZD7 and is necessary for at least one step in its localization at the base of the hair bundle.

It is notable that the paralogs PDZD7, USH1C, and DFNB31 are all deafness genes and interact with MYO7A. Moreover, PDZD7 and DFNB31 both homodimerize and form heterodimers (*Chen et al., 2014*); USH1C homodimerizes (*Siemens et al., 2002*), but whether it can heterodimerize with PDZD7 or DFNB31 is unknown.

## Other molecules co-precipitated with MYO7A

Several other proteins are specifically co-precipitated with MYO7A; those with high enrichment and relatively high stoichiometry include MYO6, MYL4, COLEC12, ANKRD24, GIPC3, and MYL1. Although each of these will require in-depth characterization such as we have done here for PDZD7, we can speculate to their possible roles in hair cells. MYL1 and MYL4 are myosin light chains, and may substitute for CALM in binding to one or more IQ domains in MYO7A (or MYO6). The presence

of MYO6 is quite interesting, as it could mediate pointed-end-directed motility in a complex that also contains a barbed-end-direct motor. More specific investigation of how MYO6 and MYO7A interact will be required; our preliminary co-expression experiments indicated only a low-stoichiometry interaction when both were expressed together in HEK cells (*Figure 5—source data 1*). A relatively large fraction of ANKRD24 present in the stereocilia-membrane extract was co-precipitated, yet ANKRD24 did not appear to bind directly to MYO7A (*Figures 5,6*). Presumably ANKRD24 interacts with MYO7A via other proteins, indicating that a larger MYO7A complex may be present that warrants further study. Other proteins were highly enriched but less abundant, including MYO3B, MYO1H, ESPN, LMO7, FSCN2, and GRXCR1. All of these proteins interact with the actin cytoskeleton, and it is likely that some of them co-precipitate because of their association with substoichiometric levels of actin filaments also bound to MYO7A. Future experiments will be needed to discern which of these proteins interacts specifically with MYO7A complexes and which co-precipitate on associated actin.

A fraction of the MYO7A derived from enriched membranes was bound to USH1C and USH1G, and likely had bound CDH23 as well. While our experiments were not designed to accurately measure stoichiometry of MYO7A complexes, the data were nonetheless consistent with a roughly equimolar complex of the three proteins. This MYO7A-USH1C-USH1G complex was not necessarily the tip-link complex itself, as MYO7A could simply transport the scaffolding proteins to the correct location at stereocilia tips, hand off the complex to CDH23, then dissociate. Notably, CDH23 does not depend on MYO7A for localization in hair bundles (*Senften et al., 2006*; *Boëda et al., 2002*), although USH1C does (*Boëda et al., 2002*; *Lëfevre et al., 2008*), suggesting that the cadherin and scaffolding proteins are transported separately. This interpretation is consistent with our determination that only a very small fraction of CDH23 was in a complex with MYO7A.

## MYO7A function in stereocilia

Unconventional myosins like MYO7A participate in transport, anchoring, tension sensing, actin organization, and cell adhesion (*Hartman et al., 2011*). Given the substantial number of scaffolding proteins found with MYO7A, it is reasonable to conclude that the motor transports various proteins towards stereocilia tips. Nevertheless, consistent localization at ankle links and tip links suggests that MYO7A may also anchor protein complexes and sense tension applied to them. Furthermore, examination of $Myo7a^{8J/8J}$ hair bundles indicates clearly that both their actin organization and stereocilia adhesion are disrupted. The division of MYO7A's tasks into several categories is thus misleading, as those activities likely all overlap—MYO7A is a multifunctional protein, interacting with several different protein complexes in stereocilia and carrying out transport and functional roles. Our hypothesis is that knowing the composition of MYO7A complexes is an essential step in understanding the function of this motor in stereocilia.

Our experiments are designed to provide a rough estimate of the abundance of proteins present in MYO7A complexes. To accurately measure the stoichiometry of the detected complexes, additional purification will be required. We can identify co-precipitating proteins using enrichment analysis, but despite a nearly 10,000-fold purification of MYO7A, the immunoaffinity eluates have many proteins that have bound nonspecifically. This result is entirely expected; the proteins that interact with MYO7A are rare within stereocilia, so MYO7A protein complexes are present at extremely small levels in the stereocilia detergent extract. In addition, because of their low levels, extensive washing of MYO7A immunoprecipitates is required, biasing our analysis towards high-affinity interactions. Future studies that use a second antibody purification step following precipitation with 138-1, for example with an anti-PDZD7 antibody, will be needed to generate a more pure complex that can be interrogated for additional binding partners and stoichiometries of each component. Such experiments will likely require further scaling up of the starting material, given the likely losses during purification. Nevertheless, there is great value in large-scale purification of protein complexes from stereocilia, as these experiments will allow us to identify interacting components—like those in the transduction-channel complex—that may not be detected by any other strategies.

## Materials and methods

### Nomenclature

All proteins are referred to by their official gene symbols, all capitals (http://www.uniprot.org; http://www.genecards.org; http://www.informatics.jax.org/mgihome/nomen/gene.shtml#ps). Actin is the principal exception, as stereocilia contain a mixture of actin isoforms that are not readily distinguished by mass spectrometry. When the mouse gene name differed from gene names in other species (e.g., *Whrn* vs. *DFNB31*), we chose the more systematic name for the protein (DFNB31 in this example).

### Antibody reagents

Hybridoma cells for the D10 monoclonal antibody were obtained from Guy Richardson (University of Sussex, UK); cells for the 138-1 monoclonal (RRID:AB_2282417) were obtained from Developmental Studies Hybridoma Bank (Iowa City, Iowa). Antibodies were purified in serum-free medium using a bioreactor (VGTI Monoclonal Core, OHSU). The control antibody for immunoaffinity purification was ChromPure Mouse IgG, whole molecule (Jackson ImmunoResearch, West Grove, PA; #015-000-003; RRID:AB_2337188).

Primary antibodies for immunoblotting or immunocytochemistry included mouse monoclonal anti-PDZD7 (AbFrontier, Seoul, Korea; #YF-PA20973, 1:500), rabbit anti-MYO7A (Proteus, Ramona, CA; #25-6790; RRID:AB_10015251), rabbit anti-ADGRV1 (from Dominick Cosgrove), and F2a rabbit anti-ATP2B2 (*Dumont et al., 2001*).

### Plasmids

The position of the fluorescent protein in the symbol for each fusion protein corresponds to the location (N- or C-terminus) relative to the stereocilia protein. The mouse GFP-MYO7A plasmid, constructed in pEGFP-C1, was obtained from David Corey. To generate the human HA-MYO7A construct, a MGC premier human MYO7A ORF clone (BC172349) was purchased from TransOMIC (Huntsville, AL; Catalog# TOH6003), amplified, and directionally cloned into pCMV6-AN-HA vector (OriGene, Rockville, MD; Catalog# PS100013) using AscI and NotI restriction enzymes. mCherry-PDZD7, mCherry-USH1C, and mCherry-USH1G, where mouse *Pdzd7*, mouse *Ush1c* (isoform b1), or human *Ush1g* were cloned into pmCherry-C1, were described previously (*Grati et al., 2012*). To make PDZD7-Strep, where 'Strep' refers to the tandem Strep tags called Strep-tag II (*Schmidt et al., 2013*), mouse *Pdzd7* was cloned into a modified pRP[Exp]-CMV vector (Vector-Builder; Santa Clara, CA) containing the tag at the 3' end of the insertion site. Mouse *Pdzd7* was also cloned into the pEGFP-N1 vector, which had been modified to remove the EGFP and add an HA tag, to generate HA-PDZD7. Mouse ANKRD24-GFP was constructed by VectorBuilder in the pRP[Exp]-CMV vector.

### Sample size and replicates

For large-scale protein purification experiments of *Figures 2–4*, the large numbers of chicken ears made pre-experiment power analysis impractical. However, the stochastic sampling that plagues shotgun mass spectrometry experiments (*Figures 2–3*) suggested that reliable detection of low abundance proteins would require multiple mass spectrometry runs, each a technical replicate. For the purification analysis (*Figure 2*), technical replicates are aliquots of the purification fractions; for the immunoaffinity purification experiments (*Figure 3*), technical replicates are divided 138-1 eluates. For practical reasons, for each experiment, we carried out two to three preparations (biological replicates) of ~500 chicken ears each. In the targeted proteomics experiments, not all proteins (and their peptides) were analyzed in each experiment (biological replicate). We ensured that each peptide analyzed in *Figure 4* was examined in at least two technical replicates from each of two biological replicates, for a total of four mass spectrometry runs each.

For the immunoprecipitation and mass spectrometry experiments of *Figure 5D–F*, which reproduce the immunoprecipitation and immunoblotting experiment of *Figure 5A* (which itself was repeated >5 times), we prepared a single biological replicate (pooled from multiple plates), and divided the 138-1 eluate so that two identical technical replicates could each be analyzed by shotgun

mass spectrometry. Although the small number of replicates precluded meaningful statistical analysis, the results qualitatively corroborated the immunoblotting experiments.

## Statistical analyses

To test whether any proteins were differentially expressed in shotgun proteomics experiments of *Figure 3*, each protein's riBAQ value was transformed into log2 scale and normalized by global median normalization (*Yang et al., 2002*). Statistical significance between conditions was determined using a modified two-sided t-test by empirical Bayes (*Smyth, 2004*), with false-discovery-rate adjustment of p-values for a multiple test correction (*Benjamini and Hochberg, 1995*). Rather than filter proteins by the number of identifications per condition, we kept as many proteins as possible as the model could be fitted. The computation was done using the limma package (*Ritchie et al., 2015*) in R Statistical Computing Environment (www.r-project.org).

In experiments that used targeted proteomics to determine whether a protein was specifically immunoprecipitated by the 138-1 antibody as compare to a control antibody, we used the two-tailed Student's t-test to compare a peptide's signal in the 138-1 eluates with that in the control eluates. In each case, 4–6 replicates from 138-1 and control eluates were compared.

## Preparation of D10 antibody beads

For immunoaffinity isolation, we used MyOne Tosylactivated Dynabeads (Life Technologies, Grand Island, NY; #65502), which are based on 1 µm superparamagnetic beads. Antibodies were coupled at 40 µg antibody per mg of beads in 0.1 M sodium borate pH 9.5, 1 M ammonium sulfate; coupling went overnight at 37°C with shaking. Unreacted groups were blocked overnight at 37°C with shaking in PBS containing 0.05% Tween 20, and 0.5% BSA. Antibody-coupled beads were stored at 4°C in the same buffer with 0.02% $NaN_3$. The D10 bead stock concentration was 50 mg/ml, with the coupled antibody at 2 mg/ml.

## Stereocilia membrane enrichment

Fertilized chicken eggs were obtained from AA Lab Eggs (Westminster, CA). Temporal bones were removed from E19-E21 chicks and were placed in ice-cold oxygenated chicken saline (155 mM NaCl, 6 mM KCl, 2 mM $MgCl_2$, 4 mM $CaCl_2$, 3 mM D-glucose, 10 mM HEPES, pH 7.25) for no more than 2 hr, with an exchange of saline after 1 hr. Sensory inner ear organs were removed using micro-dissection and were stored in ice-cold oxygenated saline for up to 4 hr during dissection. Organs were rinsed with 4–5 changes of chicken saline (minimum 10-fold dilution per rinse) to remove excess soluble protein. 3,3'-dithiobis(sulfosuccinimidyl propionate) (DTSSP; Life Technologies #21578) was added at 1 mg/ml in chicken saline for 1 hr at 4°C; organs were subjected to rotation to permit maximal access of DTSSP to extracellular spaces. To quench the DTSSP reaction, 1 ml of 1 M Tris pH 8 was added and samples were incubated for 5 min. All excess buffer was removed, then the organs were frozen by plunging into liquid N2; organs were stored at -80°C prior to use. Organs were thawed with chicken saline containing 1:100 Protease Inhibitor Cocktail (Sigma-Aldrich, St. Louis MO; #P8340) and 2% normal donkey serum (NDS; Jackson ImmunoResearch) at approximately 5 ml per 100 ears. The organs were homogenized with 20 strokes at 2400 rpm using a glass/Teflon homogenizer. The homogenate was centrifuged at 120 ×g for 5 min at 4°C. The supernatant was collected, and then the homogenization was carried out two more times. The pellet was subsequently washed 2–3 more times with chicken saline containing NDS and protease inhibitors. All supernatants (typically 50–60 ml per 1000 ears) were combined as the post-nuclear supernatant (S1); the nuclear pellet (P1) was discarded.

The post-nuclear supernatant was layered on to 2.2 M sucrose cushions (11 ml supernatant, 1 ml cushion). The samples were spun at 8400 ×g for 30 min at 4°C. The supernatant was removed (S2); to collect the dense-membrane pellet, the cushion was removed and the tubes were washed out with chicken saline with protease inhibitors and serum. Dense membranes (P2) were homogenized using five strokes in a glass/Teflon homogenizer to remove lumps. The volume yield was usually ~20–25 ml for 500 ears.

D10 beads (12.5 mg/ml beads, D10 at 500 µg/ml) were washed with chicken saline with serum and were added to the dense membranes at 1 µl per ear: the mixture was rotated overnight at 4°C. The beads were then collected with a magnet, then washed 5x with chicken saline containing serum

and 3x with chicken saline; they were then frozen while the next aliquot was prepared. The original flow-through solution was bound to fresh D10 beads using the same conditions; the next day, the beads were collected with a magnet, washed, and pooled with the original beads.

To elute stereocilia membranes, the pooled and washed D10 beads were sonicated using a sonicator (Sonics & Materials, Newtown, CT; model VCX 130) with a 2 mm probe; sonication was carried out in saline with protease inhibitors in 2–3 ml batches (in ice water to keep cold). Sonication was for 5–10 s at 25–50% power, followed by cooling in ice water for 1–2 min. A magnet was used to concentrate the beads and the solution was removed. The sonication was repeated for a total of 20 ml of eluate; this solution was spun at 112,500 $\times$g ($r_{max}$; 35,000 rpm in a Beckman 70Ti rotor); the pellet was retained. Sonication was repeated on the D10 beads with 6 $\times$ 3 ml additional aliquots; these aliquots were pooled and centrifuged. The supernatants from the two centrifugation steps were pooled (cytosolic fraction).

Membrane pellets were resuspended using sonication with saline plus protease inhibitors and were combined; the pool was diluted to ~500 ear-equivalents per tube. The solution was spun at 125,000 $\times$g ($r_{max}$; 45,000 rpm in Beckman TLA55 rotor) for 30 min at 4°C. The supernatant (S7) was removed and the pellet (enriched stereocilia membranes) was frozen at −80°C.

Stereocilia membranes were sonicated with 500 µl RIPA buffer (50 mM Tris pH 8.0, 150 mM NaCl, 0.1% SDS, 1% NP-40, 0.5% deoxycholate, 1:100 protease inhibitors) as above for each 500 ears; extracts were spun at 125,000 $\times$g ($r_{max}$) for 15 min at 4°C. The extraction was repeated twice on the pellet and the three supernatants were combined and diluted to 1.5 ml total volume (10 ears/30 µl). Aliquots used for immunoblots were precipitated with TCA and stored for later use.

## Immunoaffinity purification

Immunoaffinity purification was carried out serially; the RIPA extract was first incubated with beads with control antibody, then the unbound material was then incubated with beads coupled with the specific antibody (e.g., 138-1 anti-MYO7A). The RIPA extract (1.5 ml; 500 ear-equivalents) or flow-through material was added to 50 µl antibody-coupled beads; the beads and extract were rotated for 1 hr at room temperature. Beads were collected with a magnet, washed at least 5x with RIPA buffer, and eluted 5x with 20 µl 2% SDS. All eluates were combined for each set; typically, 20 µl was reserved for gel analysis, and 80 µl was used for mass spectrometry.

## Protein immunoblotting

Samples from the enrichment scheme were diluted to 0.05 or 0.2 ear-equivalents per 20 µl by calculating the ear number input and total volume of each fraction. NuPAGE 4X LDS sample buffer (Life Technologies #NP0008), and 500 mM DTT were added to achieve 1X LDS sample buffer and 50 mM DTT. Samples were boiled at 95°C for 2–5 min, and resolved using 4–12% SDS-PAGE gels with MES or MOPS buffer, or 3–8% SDS-PAGE gels with acetate buffer (NuPAGE gels and buffers, Life Technologies). Proteins were transferred to PVDF (Immobilon-P; EMD Millipore, Billerica MA), and were visualized with India Ink (1:5000) in PBS/0.05% Tween-20. Membranes were blocked with Prime Blocking Agent (GE Healthcare Life Sciences, Pittsburgh PA; #RPN418), and probed with specific primary antibodies which were detected with species-specific HRP-coupled secondary antibodies (Jackson ImmunoResearch #111-035-144 and #115-035-003) and ECL Prime (GE Healthcare Life Sciences #RPN2232).

## Shotgun mass spectrometry

In-solution tryptic digests of the samples were prepared using the enhanced filter-aided sample preparation (eFASP) method (*Erde et al., 2014*). Proteins were digested in an Amicon Ultra 0.5 ml 30K filter unit (Millipore; UFC503056) in 100 µl digestion buffer with 200 ng sequencing-grade modified trypsin (Promega, Madison, WI; #V5111) at 37°C for 12–16 hr. Peptides were isolated by centrifugation and were extracted with ethyl acetate to remove remaining deoxycholic acid (*Erde et al., 2014*).

Peptide samples were analyzed with an Orbitrap Fusion Tribrid mass spectrometer (Thermo Fisher Scientific, Waltham, MA) coupled to a Thermo/Dionex Ultimate 3000 Rapid Separation UPLC system and EasySpray nanosource. The samples were loaded onto an Acclaim PepMap C18, 5 µm particle, 100 µm $\times$ 2 cm trap using a 5 µl/min flow rate and then separated on an EasySpray

PepMap RSLC, C18, 2 µm particle, 75 µm × 25 cm column at a 300 nl/min flow rate. Solvent A was water and solvent B was acetonitrile, each containing 0.1% (v/v) formic acid. After loading at 2% B for 5 min, peptides were separated using a 205-min gradient from 7.5–30% B, 5-min gradient from 30–90% B, 6-min at 90% B, followed by a 19 min re-equilibration at 2% B.

MaxQuant (*Cox and Mann, 2008*) and the search engine Andromeda (*Cox et al., 2011*) were used to identify peptides and assemble proteins from the mass spectrometer RAW files. MaxQuant was used to calculate iBAQ (*Schwanhäusser et al., 2011*) for each protein, and we used an Excel spreadsheet to calculate riBAQ (*Shin et al., 2013*; *Krey et al., 2014*) and enrichment values.

Mass spectrometry data, as well as spreadsheets with all derived values, are available from ProteomeXchange (http://www.proteomexchange.org) using the accession numbers PXD004222 ("D10 Stereocilia Membrane Enrichment"), PXD004221 ("Myosin VIIA 138-1 immunoaffinity purification"), and PXD004266 ("HEK cell MYO7A-PDZD7 immunoprecipitation"); information conforming to Minimal Information About a Proteomics Experiment (MIAPE) standards (*Taylor et al., 2007*) is included in the submissions.

For clustering with mclust (*Fraley and Raftery, 2002*), shotgun riBAQ data from the D10 enrichment procedure were used if the protein was detected in at least one of the technical replicates of both biological replicates; the 2–4 measurements were then averaged. Only those proteins detected in all samples analyzed (S1, S2, P2, S3, M3, S6, P6, P7, and S7) were subjected to clustering.

## Targeted mass spectrometry

For targeted MS/MS, we measured peptides from the same three immunoprecipitations, each of ~500 ear-equivalents of stereocilia, as were used for the shotgun analysis. In-solution tryptic digests of the samples were prepared using the eFASP method. Proteins were digested in the filter unit in 100 µl digestion buffer with 200 ng sequencing-grade modified trypsin at 37°C for 12–16 hr. After isolating peptides by centrifugation, we extracted them with ethyl acetate to remove remaining deoxycholic acid (*Erde et al., 2014*). We obtained heavy peptide standards (Spiketides_L) from JPT Peptide Technologies (Berlin, Germany) for demonstration that the monitored transitions originated from the intended peptide.

Peptide samples were analyzed with an Orbitrap Fusion Tribrid mass spectrometer configured as described above; samples were loaded onto the trap and separated as above. Solvent A was water and solvent B was acetonitrile, each containing 0.1% (v/v) formic acid. After loading at 2% B for 5 min, peptides were separated using a 55-min gradient from 7.5–30% B, 10-min gradient from 30–90% B, 6-min at 90% B, followed by a 19 min re-equilibration at 2% B. Peptides were analyzed using the targeted MS2 mode of the Xcalibur software in which the doubly or triply charged precursor ion corresponding to each peptide was isolated in the quadrupole, fragmented by HCD, and full m/z 350–1600 scans of fragment ions at 30,000 resolution collected in the Orbitrap. Targeted MS2 parameters included an isolation width of 2 *m/z* for each precursor of interest, collision energy of 30%, AGC target of $5 \times 10^4$, maximum ion injection time of 100 ms, spray voltage of 2400 V, and ion transfer temperature of 275°C. No more than 75 precursors were targeted in each run and no scheduling was used. Three unique peptides for each protein of interest were chosen for isolation based on previous data-dependent discovery data or from online peptide databases (www.peptideatlas.org, www.thegpm.org). Precursor isolation lists for all peptides of interest were exported from the software package Skyline (http://proteome.gs.washington.edu/software/ skyline/) and imported into the Orbitrap control software.

Skyline was used to analyze targeted MS/MS data. Chromatographic and spectral data from RAW files were loaded into Skyline and manually analyzed to determine fragment ion peaks corresponding to each peptide. RAW files were also processed using Proteome Discoverer (Thermo Fisher Scientific) software in order to match MS/MS spectra to an Ensembl spectral database using Sequest HT. Fragment ion peaks for each peptide were chosen according to the following criteria: 1) three or more co-eluting fragment ions contributed to the peak signal, 2) at least two or more data points were collected across the peak, and 3) one or more spectra within the peak were matched to correct peptide sequence within the spectral database. If spectra within a specific sample were not identified then a) the retention time of the chosen peak must be within 2 min of the retention time of an identified peak for that peptide from another sample and b) the type of daughter ions contributing to the peak must match the identified peptide peak from another sample. If no peak matching these criteria was found in a particular sample the peak area was counted

as zero. Chromatographic peak areas from all detected fragment ions for each peptide were integrated and summed to give a final peptide peak area. Because heavy peptide standards were not used in all targeted experiments, the intensities for each sample were normalized by the relative MYO7A 138-1 eluate intensities for each experiment (*Figure 4—source data 1*). Briefly, for each MYO7A peptide, the intensity for a given experiment was divided by the average of all experiments for that peptide. These peptide normalization factors were then averaged across all MYO7A peptides, yielding an experiment normalization factor. The area for each peptide analyzed was then divided by the appropriate experiment normalization factor. The normalized peak areas were then averaged for the 4–6 replicates, and divided by the average of all S7 starting material (total) experiments. This calculation yielded the IP/total value, which was displayed for each peptide in *Figure 4*. We then used the Student's t-test to determine whether the results from the 138-1 experiments were different from the control experiments. All calculations, including the statistical analysis, are tabulated in *Figure 4—source data 1*. All targeted mass-spectrometry data are available at: https://panoramaweb.org/labkey/project/OHSU%20-%20Barr-Gillespie%20Lab/MYO7A%20PDZD7%20complex/begin.view.

## Immunoprecipitation from HEK cells

HEK293 cells (RRID:CVCL_0045) were maintained in Dulbecco's Modified Eagle Medium (DMEM; Thermo Fisher Scientific) with glucose, pyruvate, and glutamine (Sigma-Aldrich D6429), supplemented with 10% FBS (Atlanta Biologicals, Norcross, GA; #S11150). Cells were seeded 24 hr before transfection in 6 well plates at $2 \times 10^5$ cells per well. Cells were transfected with 0.4 µg DNA using Effectene (Qiagen, Hilden, Germany), then were harvested 48 hr later by removing medium, and freezing at -80°C. Proteins were extracted by agitating the plates with two washes of 500 µl RIPA buffer plus protease inhibitors (Sigma-Aldrich #P8340). The extract was cleared by centrifugation at 125,000 $\times$g ($r_{max}$; 45,000 rpm in TLA55) for 15 min at 4°C. Immunoprecipitations were carried out using 100 µl of extract and 20 µl of 138-1 beads (in PBS, 0.05% Tween-20, and 0.5% BSA at 25 mg/ml with coupled antibody at 1 mg/ml). The beads and extract were rotated for 1 hr at room temperature, collected with a magnet, washed at least 5x with RIPA, and eluted 3x with 40 µl 2% SDS. All eluates were combined; 80 µl of sample buffer with DTT was added to bring the volume to 200 µl. Total extracts (200 µl) were prepared from 100 µl of extract and 100 µl of 2x sample buffer; totals were diluted where necessary to achieve comparable signals to immunoprecipitates.

## Nanoscale pull-down assay for MYO7A partners

The nanoscale pull-down assay was conducted in COS7 cells (RRID:CVCL_0224) as described (*Bird et al., 2016*). Plasmid constructs encoding the GFP-tagged motor, IQ, and neck domains of bovine MYO10 (construct HMM-M10) was a gift from Richard Cheney (*Berg and Cheney, 2002*). The full-length cDNA clone (TOH6003) encoding human MYO7A isoform 2 (NP_001120652.1) was purchased from TransOMIC technologies (Huntsville, AL). Complementary DNA sequence encoding the human MYO7A tail, encompassing MyTH4-1, FERM-1, MyTH4-2, and FERM-2 domains (the 1208 C-terminal amino acids), was introduced into cDNA clone TOH6003 and fused to the HMM-M10 construct using EcoR I. Plasmid clones encoding the chimeric protein were selected and fully sequenced; clone M107 was used in this study.

COS7 cells were transfected using JetPrime transfection reagent (Mirus Bio LLC, Madison, WI) at 30–50% confluence. Immunocytochemistry was performed as described (*Grati and Kachar, 2011*) using a Zeiss LSM710 confocal microscope equipped with a 63X, 1.4 NA objective.

## Immunocytochemistry in inner ear

Mouse strains were C57BL/6 (RRID:IMSR_JAX:000664), as well as heterozygote and homozygote *Myo7a8J* (RRID:MGI:2182923). Inner ears from postnatal day eight (P8) mice were removed and fixed for 3 hr in 4% paraformaldehyde (in PBS). Ears were rinsed with PBS, and utricles were incubated in 50 µg/ml protease XXVII (Sigma) for 10 min to remove otolithic membranes. Organ of Corti tissues were dissected from the inner ears and the lateral wall was removed. Organs were rinsed 2 times in PBS, permeabilized for 10 min in 0.5% Triton-X in PBS, and blocked for 1 hr in PBS containing 5% normal donkey serum and 1% bovine serum albumin. Utricles were incubated overnight at 4°C in blocking solution containing primary antibodies. After three 5–10 min washes in PBS, utricles

were incubated for 3 hr at room temperature in blocking solution containing donkey anti-mouse IgG AlexaFluor 488 (Invitrogen, 1:1000) or donkey anti-rabbit IgG AlexaFluor 488 (Invitrogen, 1:1000) secondary antibodies and CF633 Phalloidin (Biotium, Fremont, CA; 1:500). Utricles were washed 3 times for 10 min each in PBS, post-fixed in 2% paraformaldehyde in PBS for 5 min, rinsed twice with PBS and then mounted with VECTASHIELD (Vector Laboratories, Burlingame, CA) on slides with 0.12 mm Secure-Seal spacers (Thermo Fisher Scientific #S-24737).

Cochlea images were acquired using a 100X, 1.46 NA Plan-Apochromat objective on a Zeiss Elyra PS.1 system that reconstructs super-resolution images from a series of images acquired under spatially structured illumination (SIM) (*Gustafsson, 2000*). Images were processed for SIM reconstruction in Zen 2012 (Zeiss) and three-dimensional projections of each z-stack were visualized and converted into TIFF files using the 'Surpass' mode and 'Snapshot' tool on Biplane Imaris 7.

Utricle images were acquired using either standard confocal microscopy or Airyscan processing on a Zeiss LSM780 or LSM880 with Airyscan using a Plan-Apochromat 63X, 1.4 NA objective. Airyscan processing was carried out using Zen 2012 (Zeiss). All z-stacks were processed using Bitplane Imaris 7. Three dimensional projections of each z-stack were visualized using the 'Surpass' mode and were cropped along the x or y dimension to include one row of hair cells using the 'Crop 3D' tool. The cropped 3D projection was then rotated in Surpass mode to visualize a y-z or x-z slab of the stack with the hair bundles oriented vertically and the 'Snapshot' tool was used to generate a tiff image of the projection.

## Stereocilia isolation on glass

Utricles were dissected and otoconia were removed with an eyelash. Square #1.5 glass coverslips (Corning) were washed with water and 70% ethanol, then autoclaved; they were then coated with 100 μg/ml poly-L-lysine for 10–20 min. Poly-L-lysine was removed and coverslips dried for 30–60 min. Dissected utricles were dropped onto the coverslips in dissection media and gently pressed to the coverslip with the hair bundle side down. The utricle and dissection solution was then removed and 200 μl of 4% formaldehyde (Electron Microscopy Sciences) in PBS was added to each coverslip for 20 min. Coverslips were rinsed three times with PBS and then placed on top of a petri dish lid inside a humidity chamber. Samples were permeabilized for 15 min in 0.2% Triton X-100 and 2% normal donkey serum in PBS, and blocked for 1 hr in 5% normal donkey serum in PBS. Coverslips were incubated overnight at 4°C with primary antibodies diluted in blocking solution, then rinsed 3x for 10 min each. Coverslips were then incubated for 3–4 hr in blocking solution with donkey Alexa-Fluor 488, 568 and/or 647 (1:1000, Invitrogen) secondary antibodies (unless primary antibody was directly labeled) and 0.4 U/ml Alexa Fluor 568 Phalloidin (Molecular Probes, Invitrogen) or CF633 Phalloidin (Biotium, 1:500) followed by three 10-min rinses with PBS. Coverslips were rinsed briefly in water; excess water was blotted on a paper towel, then coverslips were mounted on slides using EverBrite mounting medium (Biotium).

Isolated stereocilia images were acquired using a 100X, 1.46 NA Plan-Apochromat objective on the Zeiss Elyra PS.1 system. Images were processed for SIM reconstruction in Zen 2012 (Zeiss) and selected Z-planes were exported as tiff images in Biplane Imaris 7. For the analysis in *Figure 8E*, the line tool was used in Fiji to draw a line from the taper of each indicated stereocilium to its tip; the line was approximately the width of the stereocilia shafts. 'Plot Profile' was used to determine the pixel intensity along each stereocilium. In Excel, the profiles were aligned at the tapers, then all 11 profiles were averaged. Because each stereocilium is of a different length, only the taper and shaft regions are in alignment.

## In utero electroporation

In utero electroporation was carried out as described previously (*Ebrahim et al., 2016*). Briefly, C57BL/6 males were crossed to CD1 females to generate embryos for transuterine microinjection. Pregnant females with E11.5 embryos were anesthetized with Nembutal (60–65 μg/gram body weight) in a 20.8 mg/ml MgSO4, 10% ethanol, and 40% propylene glycol. The uterine horn was exposed using a ventral laparotomy and embryos were visualized and positioned using a low-intensity halogen light. A microinjection pipette, backfilled with concentrated DNA plasmid (>3 μg/μl), was secured in place in a pipette holder coupled to a Picospritzer III microinjector. The micropipette was inserted into the otocyst, and compressed nitrogen was used to deliver the inoculum. A square

wave electroporator (Protech International CUY21SC) was then used to deliver a pulse of 60–100 mA to the injected embryo. Once the transuterine and electroporation steps were complete, the abdominal wall was closed using an absorbable suture. Immunocytochemistry was carried out as described above.

## Injectoporation in inner ear

Experiments were performed essentially as described previously (*Xiong et al., 2012*; *Zhao et al., 2014*). The organ of Corti was isolated from P3 mice and placed in DMEM/F12 medium with 1.5 µg/ml ampicillin. For electroporation, glass electrodes (2 µm diameter) were used to deliver plasmid (500 ng/µl in 1x HBSS) to the sensory epithelium. A series of three pulses was applied; pulses were at 1 sec intervals with a magnitude of 60 V and duration of 15 msec (ECM 830 square wave electroporator; BTX). The tissue was fixed after two days in culture and stained with phalloidin; the mCherry signal was directly imaged for PDZD7.

## Acknowledgements

We received support from the following core facilities: mass spectrometry from the OHSU Proteomics Shared Resource (partial support from NIH core grants P30 EY010572 & P30 CA069533; Orbitrap Fusion S10 OD012246), hybridoma cell culture and antibody production from the VGTI Monoclonal Antibody Core (OHSU), and confocal microscopy from the OHSU Advanced Light Microscopy Core @ The Jungers Center. The 138-1 hybridoma was developed by DJ Orten and T Hasson and was obtained from the Developmental Studies Hybridoma Bank, created by the NICHD of the NIH and maintained at The University of Iowa, Department of Biology, Iowa City, IA 52242.

## Additional information

### Funding

| Funder | Grant reference number | Author |
| --- | --- | --- |
| National Institute on Deafness and Other Communication Disorders | F32DC012455 | Jocelyn F Krey |
| National Institute on Deafness and Other Communication Disorders | R03DC014544 | Jocelyn F Krey |
| National Institute on Deafness and Other Communication Disorders | R01DC05575 | Xue Zhong Liu |
| National Institute on Deafness and Other Communication Disorders | R01DC012546 | Xue Zhong Liu |
| National Institute on Deafness and Other Communication Disorders | R01DC012115 | Xue Zhong Liu |
| National Institute on Deafness and Other Communication Disorders | R01DC014427 | Ulrich Müller Peter G Barr-Gillespie |
| National Institute on Deafness and Other Communication Disorders | R01DC005965 | Ulrich Müller |
| National Institute on Deafness and Other Communication Disorders | R01DC002368 | Peter G Barr-Gillespie |
| National Institute on Deafness and Other Communication Disorders | P30DC005983 | Peter G Barr-Gillespie |

The funders had no role in study design, data collection and interpretation, or the decision to submit the work for publication.

## Author contributions
CPM, JFK, MG, PGB-G, Conception and design, Acquisition of data, Analysis and interpretation of data, Drafting or revising the article; BZ, SF, AK-S, Acquisition of data, Analysis and interpretation of data, Drafting or revising the article; XZL, DC, UM, Conception and design, Analysis and interpretation of data, Drafting or revising the article

## Author ORCIDs
Peter G Barr-Gillespie, iD http://orcid.org/0000-0002-9787-5860

## Ethics
Animal experimentation: This study was performed in strict accordance with the recommendations in the Guide for the Care and Use of Laboratory Animals of the National Institutes of Health. All of the animals were handled and euthanized according to a protocol (#IS00003292) that was approved by the institutional animal care and use committee (IACUC) of the Oregon Health & Science University.

# Additional files

## Supplementary files
• Reporting standards 1. Minimum Information About a Proteomics Experiment guidlines and parameters (GIL160, GIL167).

• Reporting standards 2. Minimum Information About a Proteomics Experiment guidlines and parameters (GIL1467, GIL1649, GIL1651).

• Reporting standards 3. Minimum Information About a Proteomics Experiment guidlines and parameters (GILL-85, GILL-98, GILL-102).

## Major datasets
The following datasets were generated:

| Author(s) | Year | Dataset title | Dataset URL | Database, license, and accessibility information |
|---|---|---|---|---|
| Barr-Gillespie PG | 2016 | D10 Stereocilia Membrane Enrichment | www.ebi.ac.uk/pride/archive/projects/PXD004222 | Available at the PRIDE Archive (accession no. PXD004222) |
| Barr-Gillespie PG | 2016 | Myosin VIIA 138-1 immunoaffinity purification | www.ebi.ac.uk/pride/archive/projects/PXD004221 | Available at the PRIDE Archive (accession no. PXD004221) |
| Barr-Gillespie PG | 2016 | HEK cell MYO7A-PDZD7 immunoprecipitation | www.ebi.ac.uk/pride/archive/projects/PXD004266 | Available at the PRIDE Archive (accession no. PXD004266) |
| Krey JF | 2016 | MYO7A PDZD7 complex | https://panoramaweb.org/labkey/project/OHSU%20-%20Barr-Gillespie%20Lab/MYO7A%20PDZD7%20complex/begin.view | Publicly accessible at panoramaweb.org |

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
