## [Decision Letter]

Thank you for submitting your manuscript " PDZD7-MYO7A complex identified in enriched stereocilia membranes" to *eLife*. The manuscript has been reviewed by three expert reviewers, and their assessments together with my own, forms the basis of this letter. I am also including the three reviews in their original form at the end of this letter, as there are many specific and useful suggestions in them that will not be repeated in the summary here. The following individuals involved in review of your submission have agreed to reveal their identity: William Brownell (Reviewer #2); Anthony J Ricci (Reviewer #3).

All of the reviewers were impressed with the technical prowess of your experiments. Inner ear proteomics is not for the faint of heart.

Reviewer #1

Barr-Gillespie and colleagues have pioneered a proteomics approach to understand the molecular composition of the hair-cell stereocilia. Until now, this has revealed the total proteome but not discrete molecular complexes. Here, they use a purification protocol to cross-link proteins, pull down membranes with an antibody to an abundant membrane protein, and then specifically isolate proteins associated with MYO7A using a secondary immunoprecipitation. The paper illustrates the power of the method, but also how very difficult it is to do biochemistry at this scale. Even if few will have the courage to follow this path, the paper is an important contribution both to the technology and to the specific molecular understanding of stereocilia. As submitted, the paper is very complete with only few areas for polishing.

Results section, subsection “Immunoaffinity purification of MYO7A complexes from chick stereocilia” and Discussion: Although the DTSSP step will crosslink some proteins that are only mildly associated, it seems surprising that the 138-1 runs pull down so many proteins (MYO3B, MYO1H, MYO6, ESPN, LMO7, FSCN2, GRXCR1, GIPC3) and it seems unlikely that all are part of MYO7A complexes. The Discussion could be expanded to dwell a bit more on how to distinguish real from spurious associations.

Discussion section, subsection “Other molecules co-precipitated with MYO7A”: MYO7A would put ankle links under tension only if just one end moved. But the other end is presumably also associated with MYO7A.

Reviewer #2

The rationale for the study is that MYO7A may target the ankle link region of the stereocilia bundle because of interactions with specific proteins. The number of molecules involved in these interactions is very small and the interacting proteins would not be seen with traditional screens. The premise is that a high throughput screen of 1000s of stereociliar bundles may yield sufficient material for mass spectroscopic determination of candidate interacting proteins. The bulk of the manuscript is a presentation of the protocol used to accomplish the identification of PDZD7 as a potential interaction protein. The identification is verified using immunohistochemical labeling in culture cells and hair cells. The manuscript is essentially a methods paper with extraordinary expository power. The writing is spellbinding and held my interest throughout even though I do not totally understand the reason for each branch of the flow diagram describing the experimental design.

The rationale is somewhat slippery. Even though MYO7A is a deafness gene known to be essential for mechanotransduction, its function in the ankle links region is far from understood. A few sentences speculating on the role of MYO7A and specifically its interaction with PDZD7 would be appropriate and raise the general interest of the paper.

The following paper has recently been published. It may provide additional support in the background material on PDZD7.

Vona B, Lechno S, Hofrichter MA, Hopf S, Laig AK, Haaf T, Keilmann A, Zechner U, Bartsch O. Confirmation of PDZD7 as a Nonsyndromic Hearing Loss Gene. Ear Hear. 2016;37(4):e238-46. doi: 10.1097/AUD.0000000000000278. PubMed PMID: 26849169.

Reviewer #3

Morgan et al. provide a detailed methodology for isolating membrane targeted hair bundle proteins from chick inner ear. This method requires less surgical and physical separation of hair bundles but rather relies on pulling hair bundle stereocilia from homogenized inner ear tissue by targeting D10 antibody affinity of the stereocilia. Membrane proteins were targeted and stabilized using treatment with DTSSP a reversible crosslinker. Biochemical and proteomic methods were used to isolate, purify, enrich and identify stereocilia proteins and specifically the interacting partners of Myo7A. Overall this work provides good evidence supporting the ability of this new method to isolate stereociliary proteins as well as to investigate protein clustering. There are a couple of issues that could be addressed to make the significance of the work more clear both methodologically and biologically.

1) A more direct comparison to the twist off method would be helpful in terms of sensitivity, i.e. what is the lowest level of protein that you might identify between these technologies. What was the level of enrichment comparatively between the methods? What is the level of protein retrieval (i.e. what do you have left to work with) between the methods? What is the final purity level?

2) Have you compared the DTSSP method of obtaining membrane proteins between stereocilia isolation methods? Are they comparable?

3) I could not get my head around Figure 2, are these 25 different protein clusters? What proteins are involved? Neither the text nor the legend discusses 25 clusters, just cluster 8? What is different about each one?

4) I didn't see where Figure 4 was discussed and was also unclear of its significance?

5) Figure 7 seems to largely be confirming published data, I am not sure how important these data are to include (though I do think replicating published data is important).

6) What is rate limiting when discussing abundance of proteins, are all proteins and protein interactions equally weighted? That is, is the process to get these proteins linear or might it be biased toward higher affinity interactions or might some steps in the purification and isolation procedure selectively effect some proteins and not others?

7) It is unclear how the plots in Figure 8 were generated? How many cilia are included in measurement? How are they aligned, how do you know top from bottom and do you only look at cilia of a particular length?

8) It was surprising that Myo Ic was not identified by this methodology, does this reflect sensitivity of the assay?

No additional data is required, perhaps a bit of additional analysis. For most of the data presented the rigor and statistical analysis were appropriate. A couple of points raised to authors directly address where this might not be quite the case.

---

## [Author Response]

*Reviewer #1*

*Results section, subsection “Immunoaffinity purification of MYO7A complexes from chick stereocilia” and Discussion: Although the DTSSP step will crosslink some proteins that are only mildly associated, it seems surprising that the 138-1 runs pull down so many proteins (MYO3B, MYO1H, MYO6, ESPN, LMO7, FSCN2, GRXCR1, GIPC3) and it seems unlikely that all are part of MYO7A complexes. The Discussion could be expanded to dwell a bit more on how to distinguish real from spurious associations.*

DTSSP is a membrane-impermeant and thus should—in principle—crosslink only transmembrane and extracellular proteins. Nevertheless, membrane lysis during procedure might provide access of DTSSP to the cytoplasm, leading to crosslinking of some cytoplasmic proteins at low stoichiometry. Our experience with PDZD7 shows that each association detected will require an extensive set of experiments to validate, and so we have refrained from speculating too much on the biological significance of other proteins pulled down. Perhaps we erred on the side of too little speculation, especially (as the reviewer notes) on how to identify the real associations. Accordingly, we have modified the section to include the following text: *“Subsequent validation of interactions detected, for example showing co-immunoprecipitation, protein-protein binding, or mislocalization in mutant hair cells, is thus of high importance.”* (bold is new text) We also expanded the discussion later under the heading *“Other molecules co-precipitated with MYO7A,” adding the following text: “Other proteins were highly enriched but less abundant, including MYO3B, MYO1H, ESPN, LMO7, FSCN2, and GRXCR1. All of these proteins interact with the actin cytoskeleton, and it is likely that some of them co-precipitate because of their association with substoichiometric levels of actin filaments also bound to MYO7A. Future experiments will be needed to discern which of these proteins interacts specifically with MYO7A complexes and which co-precipitate on associated actin.”*

*Discussion section, subsection “Other molecules co-precipitated with MYO7A”: MYO7A would put ankle links under tension only if just one end moved. But the other end is presumably also associated with MYO7A.*

Yes, the point is well taken, but we did consider that in writing this section: *“MYO7A may move PDZD7 and ADGRV1 towards stereocilia tips but stall if the N-terminal, extracellular end of ADGRV1 was anchored on the adjacent stereocilium.”* Anchoring could occur if the N-terminus of ADGRV1 bound to an immobilized protein (USH2A) or to another ADGRV1 that is anchored on the C-terminus by other molecules than MYO7A (or rigor MYO7A). It is not clear whether an ankle link has a structure of MYO7A- ADGRV1-ADGRV1-MYO7A; evidence for an ADGRV1 antiparallel end-to-end dimer is not strong.

*Reviewer #2*

*The rationale is somewhat slippery. Even though MYO7A is a deafness gene known to be essential for mechanotransduction, its function in the ankle links region is far from understood. A few sentences speculating on the role of MYO7A and specifically its interaction with PDZD7 would be appropriate and raise the general interest of the paper.*

We were surprised by this comment, as the second paragraph in the Discussion section titled “Association of MYO7A with PDZD7 and other ankle-link components” speculates on the role of the MYO7A-PDZD7 interaction. However, we also assume the reviewer also wants more general discussion on the role of MYO7A. We added a new section, “MYO7A function in stereocilia”, into the Discussion, rearranging the last several paragraphs as well. The key new paragraph is as follows:

“Unconventional myosins like MYO7A participate in transport, anchoring, tension sensing, actin organization, and cell adhesion (Hartman et el., 2011). […] Our hypothesis is that knowing the composition of MYO7A complexes is an essential step in understanding the function of this motor in stereocilia.”

*The following paper has recently been published. It may provide additional support in the background material on PDZD7.*

*Vona B, Lechno S, Hofrichter MA, Hopf S, Laig AK, Haaf T, Keilmann A, Zechner U, Bartsch O. Confirmation of PDZD7 as a Nonsyndromic Hearing Loss Gene. Ear Hear. 2016;37(4):e238-46. doi: 10.1097/AUD.0000000000000278. PubMed PMID: 26849169.*

Thank you for calling our attention to this reference. The Discussion was modified to include this sentence: *“Originally identified as a modifier gene for Usher syndrome type 2 genes, PDZD7 has more recently been identified as a nonsyndromic deafness gene itself (Booth et al., 2015; Vona et al., 2016).”*

*Reviewer #3*

1) A more direct comparison to the twist off method would be helpful in terms of sensitivity, i.e. what is the lowest level of protein that you might identify between these technologies. What was the level of enrichment comparatively between the methods? What is the level of protein retrieval (i.e. what do you have left to work with) between the methods?

This is a difficult question to answer, as the two preparations are apples and oranges—one enriches stereocilia membranes, and the other isolates whole stereocilia. More importantly, mass spectrometric analysis is highly stochastic, especially for low-abundance proteins; increasing the number of runs leads to more and more proteins identified, but some proteins (e.g., membrane proteins) are more difficult to detect because of their composition. With the twist-off method, we feel that we have reached a plateau where we detect most proteins at >10 molecules per stereocilia, but detection of more rare proteins requires additional enrichment. That is one of the main goals of the present work: to provide tools for further enrichment.

To partially address the reviewer's question, we have added a new supplemental figure, Figure 1—figure supplement 1, which compares the recovery per ear, relative to the amount in a single utricle, for several proteins in the D10 and twist-off preparations. In part due to the larger number of organs sampled (7 per ear rather than just 1), much more actin and ATP2B2 is isolated in the D10 prep than in the twist-offs. Thus in one important sense, the D10 preparation yields 10x more stereocilia membrane protein per ear—and the ears/day throughput is much greater.

We have also summarized the comparison between the D10 and twist-off methods in a new table, Table 1.

What is the final purity level?

We took advantage of the mass spectrometry data to make an estimate of purity of the S7 fraction. We decided to emphasize membrane purity, using markers for stereocilia, basolateral, and mitochondrial membranes to make our estimate. Other non-membrane proteins are also present at moderate levels, but cannot be directly compared to membrane proteins, which are poorly detected and therefore under-quantified by mass spectrometry. The markers we chose have known densities in their membranes, allowing us to weight the riBAQ values to estimate membrane amount solubilized, rather than membrane protein solubilized. The calculations suggested that the amount of stereocilia membrane was ~5% of the total membrane solubilized by RIPA. The appropriate text in the Results was:

“The most abundant membrane proteins in S7 were either from plasma membrane (e.g., ATP1A1, at riBAQ = 1.1 x 10^-2^) or mitochondrial membranes (e.g., ATP5A1, at riBAQ = 1.1 x 10^-2^). In comparison, ATP2B2, marking stereocilia membranes, was present at riBAQ = 0.7 x 10^-2^. In frog stereocilia, ATP2B2 and ATP1A1 each have membrane densities of ~3000 µm^-2^ (Yamoah et al., 1998; Burnham and Stirling, 1984); in rat liver mitochondria, ATP5A1 has a density of ~7500 µm^-2^ (Schwerzmann et al., 1986). Taken together, these data suggest that stereocilia membranes account for ~5% of the solubilized membrane material in S7.”

We also rewrote a couple of sentences in the Discussion: “The D10 preparation has significant contamination, however; we estimate that stereocilia membranes account for only ~5% of the RIPA-solubilized material. Other membranes are present in part because contaminating intracellular organelles are disrupted by a freeze-thaw cycle, preventing their precise separation from stereocilia based on distinct, uniform sedimentation properties (Cox and Emili, 2006).”

*2) Have you compared the DTSSP method of obtaining membrane proteins between stereocilia isolation methods? Are they comparable?*

We were not quite sure what the reviewer is asking—(1) whether DTSSP was effective in stabilizing membrane protein complexes, or (2) how the D10 method (with DTSSP) compared to the twist-off method. Because the present manuscript does not focus on membrane proteins, we are leaving the data supporting point #1 (showing more PCDH15 co-immunoprecipitates with TMC1 if the tissue is pretreated with DTSSP) for a subsequent manuscript. As to point #2, it is a good question that is addressed by Figure 1—figure supplement 1.

3) I could not get my head around Figure 2, are these 25 different protein clusters? What proteins are involved? Neither the text nor the legend discusses 25 clusters, just cluster 8? What is different about each one?

All proteins that were detected in each of the considered purification fractions were analyzed (no missing data allowed). They are different in their patterns of protein expression—proteins with similar patterns of expression are grouped into a cluster. We found that cluster 8 was most relevant for this analysis, as it both had the steepest enrichment slope and it contained the key proteins PTPRQ and ATP2B2. We have now added a new source data file ([Supplementary-material SD2-data]), which contains the normalized expression levels for each protein in each fraction, as well as the cluster assignment. Interested readers can peruse the tables to see what proteins are clustered together.

4) I didn't see where Figure 4 was discussed and was also unclear of its significance?

Figure 4 is important—it illustrates how we sum the daughter ions (fragments of the original peptide) over time to get the total signal for the peptide of interest. The legend was changed to read: *“**C**, Each daughter ion is monitored over time; signal from all monitored daughter ions is summed and the time-summed intensity plot is integrated to determine the signal for the peptide of interest.”*

*5) Figure 7 seems to largely be confirming published data, I am not sure how important these data are to include (though I do think replicating published data is important).*

There are many new observations in Figure 7, and while the gist is confirmation of sketchy previous data, the new observations help us in understanding what MYO7A is doing. Figure 7 validates the antibody for use in the vestibular system. Figure 7 is the first SIM localization of MYO7A, highlighting its location outside the actin core. Figure 7 reveal that the mutant has (a) longer stereocilia, (b) probably fewer stereocilia, (c) disorganized stereocilia, and (d) a mislocalized fonticulus. Figure 7 confirm in the vestibular system what has been shown in the auditory system previously, and Figure 7 show that the mislocalization of ADGRV1 does not extend to another membrane protein. Finally, Figure 7 reveal much more detail of the phenotype in cochlear hair cells, as SIM gives far better resolution.

6) What is rate limiting when discussing abundance of proteins, are all proteins and protein interactions equally weighted? That is, is the process to get these proteins linear or might it be biased toward higher affinity interactions or might some steps in the purification and isolation procedure selectively effect some proteins and not others?

The affinity of the interactions in the MYO7A complex is definitely an important point, especially for the final immunoaffinity purification step. The critical issue is the wash time that follows the antibody-bead interaction with the extract; this wash step takes ~30 minutes for us. Presumably dissociation rates for different MYO7A complexes vary substantially, so some complexes may be nearly completely stable and others may be quite transient. Here, we considered the stoichiometry of each protein relative to MYO7A when choosing which to focus on; complexes with near 1:1 stoichiometry must be quite stable, assuming there is not an abundance of one of the partners in the complex.

Thus the picture of the MYO7A complexes afforded by our immunoaffinity experiments is distorted by complex affinity. These experiments must therefore be used to discover new complexes, but the detailed composition, including stoichiometries, should be probed in different experiments.

We have added the following sentence to the last paragraph of the Discussion: “In addition, because of their low levels, extensive washing of MYO7A immunoprecipitates is required, biasing our analysis towards high-affinity interactions.”

Note that it is possible to use membrane-permeant crosslinkers instead of DTTSP, which would stabilize intracellular complexes as well. We have had successful pilot experiments with these and will be pursuing this route in the future.

*7) It is unclear how the plots in Figure 8 were generated? How many cilia are included in measurement? How are they aligned, how do you know top from bottom and do you only look at cilia of a particular length?*

We added the following to the Methods: “For the analysis in Figure 8, the line tool was used in Fiji to draw a line from the taper of each indicated stereocilium to its tip; the line was approximately the width of the stereocilia shafts. "Plot Profile" was used to determine the pixel intensity along each stereocilium. In Excel, the profiles were aligned at the tapers, then all 11 profiles were averaged. Because each stereocilium is of a different length, only the taper and shaft regions are in alignment.”

*8) It was surprising that Myo Ic was not identified by this methodology, does this reflect sensitivity of the assay?*

MYO1C was identified—see Figure 3. Its enrichment was not particularly high, however, and we did not call it out in the Results.